# RACER: Retrieval-Augmented Contextual Rapid Speculative Decoding

## Abstract

Autoregressive decoding in Large Language Models (LLMs) generates one token per step, causing high inference latency. Speculative decoding (SD) mitigates this through a guess-and-verify strategy, but existing training-free variants face trade-offs: retrieval-based drafts break when no exact match exists, while logits-based drafts lack structural guidance. We propose **RACER** (**R**etrieval-**A**ugmented **C**ont**e**xtual **R**apid Speculative Decoding), a lightweight and training-free framework that integrates retrieved exact patterns with logit-driven future cues. This unification supplies both reliable anchors and flexible extrapolation, yielding richer speculative drafts. Experiments on Spec-Bench, HumanEval, and MGSM demonstrate that RACER consistently accelerates inference, achieving a speedup of $2.2 \sim 2.8 \times$ compared to autoregressive decoding, and outperforms prior training-free methods, offering a scalable, plug-and-play solution for efficient LLM decoding. Our source code is available at this anonymous repository.

## 1 Introduction

Large Language Models (LLMs) such as GPT (OpenAI, 2025), LLaMA (Dubey et al., 2024), and Qwen (Bai et al., 2023) have achieved remarkable success across diverse natural language processing tasks. However, their autoregressive decoding paradigm, which generates one token per step, fundamentally limits inference efficiency. The sequential dependency causes inference latency to scale linearly with sequence length and model size, creating a key bottleneck for real-world deployment.

Speculative Decoding (SD) has emerged as a promising approach to address this challenge. By adopting a *guess-and-verify* strategy, SD enables multiple tokens to be proposed and verified in parallel, achieving acceleration without sacrificing output quality. Existing methods fall into two categories. Model-based approaches rely on lightweight auxiliary models – either separately trained (Cai et al., 2024; Li et al., 2024a;b; 2025) or inherited from smaller variants of the same model family (Leviathan et al., 2023) – to generate draft tokens, at the cost of additional memory, training, and integration overhead. Model-free approaches, in contrast, construct draft tokens directly from signals available during inference. Among model-free approaches, most are retrieval-based, leveraging exact token sequence matches from static corpora or dynamically generated contexts (Saxena, 2023; He et al., 2023). Recent work further exploits the predictive power of last logit (Liu et al., 2025b), or recycles candidate logits (Luo et al., 2025), showing that LLMs inherently encode richer cues for near-future tokens than previously assumed.

Despite these advances in model-free methods, two key limitations remain. First, retrieval-based methods depend on exact token matching, which breaks down when no continuation can be directly aligned. Second, logits-based methods are restricted to last-step or self-drafted candidates and lack external structural guidance, making it difficult to extrapolate toward more suitable tokens. As a result, their predictions tend to be narrow in scope and suboptimal in quality.

To address these limitations, we propose **RACER** (**R**etrieval-**A**ugmented **C**ontextual **R**apid Speculative Decoding), a **plug-and-play, training-free** method that unifies the strengths of both paradigms. Retrieval provides **seen information** through exact pattern matches, offering structural guidance, while logits supply **unseen information**, enabling extrapolation beyond strict matches. By augmenting logit predictions with retrieval signals, RACER generates richer and more accurate speculative

drafts. In this way, retrieval functions not as an independent generator but as structural guidance that empowers logits to hypothesize plausible continuations beyond their immediate horizon.

We conduct comprehensive experiments among general benchmark Spec-Bench (Xia et al., 2024), code generation benchmark HumanEval (Chen et al., 2021) and Chinese math reasoning benchmark MGSM (Shi et al., 2022). Our contribution is threefold:

1. We identify and exploit the complementary nature of seen and unseen information for speculative decoding.
2. We introduce a unified cache-like framework that integrates retrieval-based draft trees and logit-based dynamic updates.
3. We show that RACER achieves superior inference acceleration with stable memory usage, establishing a lightweight and scalable foundation for training-free speculative decoding.

## 2 BACKGROUND

In this section, we provide the necessary background and related works on speculative decoding.

**Speculative Decoding** Speculative Decoding (SD) typically proceeds in two phases: a *drafting phase* and a *verification phase*.

Given a prefix $\mathbf{x}$, a lightweight **draft model** $M_q$ generates $\gamma$ candidate tokens $\tilde{x}_1, \ldots, \tilde{x}_\gamma$. These tokens, together with the prefix, are then passed to the **target model** $M_p$, which produces logits $p_1, \ldots, p_\gamma$. Each draft token $\tilde{x}_i$ is verified by comparing its probability under $M_p$ with that under $M_q$ (Leviathan et al., 2023; Chen et al., 2023):

$$\alpha_i = \begin{cases} 1 & \text{if } p_i[\tilde{x}_i] \geq q_i[\tilde{x}_i], \\ \dfrac{p_i[\tilde{x}_i]}{q_i[\tilde{x}_i]} & \text{otherwise.} \end{cases} \tag{1}$$

If $\tilde{x}_i$ is accepted (with probability $\alpha_i$), it is appended to the sequence; otherwise, $\tilde{x}_i, \ldots, \tilde{x}_\gamma$ are discarded, and the speculative decoding step terminates early. The next iteration then resumes from the last accepted prefix, using the last accepted logits $p_{i-1}$ of the target model to resample $x_i$ as the new continuation token. This guarantees that every iteration produces at least one token, while leveraging accepted drafts whenever possible to accelerate generation.

Regardless of whether greedy or nucleus sampling is employed, validation always leverages the logits from the target model. In expectation, a single SD step can advance by up to $\gamma + 1$ tokens, significantly reducing the number of target model invocations compared to standard AR decoding.

**Retrieval-based Speculative Decoding** Retrieval-based SD methods bypass the draft model $M_q$ and instead rely on pattern matching within the token sequence. PLD (Saxena, 2023), for example, stores past $n$-grams together with their succeeding $m$-grams as predictions. This method is simple and effective in pattern-repeating scenarios such as code generation, but can only propose a single continuation at a time. Moreover, because pattern matches are sparse and fail to capture the full diversity of target model outputs, PLD is constrained to specific domains and cannot generalize broadly.

**Tree Attention** The standard guess-and-verify scheme assumes a linear draft sequence. *Tree attention* (Cai et al., 2024) generalizes this by allowing the draft model to propose a branching tree of candidates. During verification, the target model processes all nodes in parallel, with position encodings set by depth and attention masks restricting each node to its ancestors:

$$\text{pos}[i] = \text{pos}[\text{parent}(i)] + 1, \quad \text{mask}[i, j] = \mathbb{1}[\, j = i \text{ or } j \in \text{ancestor}(i)\,]. \tag{2}$$

This transforms speculative decoding into a branching search process, enabling higher parallelism and more effective utilization of the target model when multiple plausible continuations exist.

For model-based methods, Medusa (Cai et al., 2024) attaches multiple additional LM heads to the top layer, each predicting draft tokens for different tree depths. EAGLE-3 (Li et al., 2025) further

integrates low-, mid-, and high-level features of the target model, with its core structured as a Transformer decoder layer. For model-free methods, REST (He et al., 2023) constructs a suffix array to identify the longest suffix match and expands the matched continuation into a trie, while SAM Decoding (Hu et al., 2025) employs both dynamic and static suffix automata to capture contextual as well as pre-built suffix patterns, providing more flexible retrieval-guided expansions.

## 3 METHODOLOGY

In this section, we introduce our approach in three parts: *Logits Tree*, *Retrieval Tree with LRU (Least Recently Used) eviction*, and their integration strategy. Logits Tree leverages predictive distributions to hypothesize unseen continuations, while Retrieval Tree reuses observed contexts to provide reliable structural guidance. Their combination yields a unified framework that balances generalization and memorization.

### 3.1 LOGITS TREE

**First Step beyond Next-Token** We examine two *logit-reuse* strategies when extending the tree beyond the next token. The *last-logit* strategy reuses the logit distribution from which the next-token was sampled to expand all of its children candidates, assuming local smoothness in the token space. The *copy-logit* strategy instead reuses the logit from the most recent occurrence of the same token, assuming that identical vocabulary tokens tend to preserve similar semantic tendencies when appearing in comparable contexts. Here, "the same token" refers to the same vocabulary token (same token ID), and we reuse the logits from its most recent occurrence in the history. To evaluate these strategies, we conducted experiments on Spec-Bench using Vicuna-7B-v1.5 and Qwen3-1.7B under greedy decoding. For each speculative step, we selected the top-63 tokens from the corresponding logits to form the first layer, and measured their effectiveness by the Mean Accepted Tokens (MAT) and the distribution of accepted ranks (from 1 to 63). Since both models exhibited similar trends, we report only the results of Vicuna-7B-v1.5 here. Additional results could be found in Appendix E.

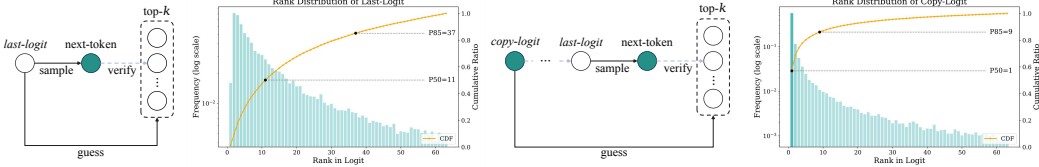

Figure 1: The *last-logit* node (white) produces both the next-token sample and the draft tokens immediately after it. The *copy-logit* node (green) marks the same token ID as the next-token, whose logit is reused to approximate the next-token's logits when generating subsequent draft tokens. After the target model digests the next-token, its logit is used to verify only one draft token in the first layer of the Logits Tree. In the histogram, the x-axis denotes the reused-logit rank of each draft token, and the y-axis reports their relative frequency among accepted draft tokens.

The MAT values of *last-logit* and *copy-logit* are 1.57 and 1.87, respectively, indicating that *copy-logit* achieves higher accepted rate. As shown in Figure 1, the *copy-logit* strategy also exhibits a pronounced heavy-tail property: its accepted tokens concentrate strongly at the top ranks, with rank-1 alone accounting for more than $50\%$ of accepted cases. For *copy-logit*, the 50th and 85th percentile accepted ranks are 1 and 9, compared with 11 and 37 for *last-logit*. These results demonstrate that *copy-logit* provides a sharper and more reliable distribution for speculative expansion, and we therefore adopt it as the basic expansion strategy of Logits Tree.

*k*-**ary Analogy and Pruning** Motivated by the heavy-tail property observed in Figure 1, we next extend the ex-

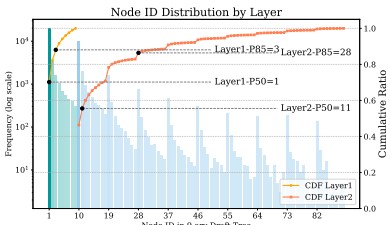

Figure 2: Analogy experiments with a fixed 9-ary draft tree of height 3, illustrating the accumulative trends across different parents and motivating the proposed breadth allocation rule for effective pruning.

pansion recursively to deeper layers. Two principles guide this design: (i) the 85th percentile rank is around 9, indicating that useful candidates concentrate in the head of the distribution; and (ii) because speculative decoding proceeds prefix-wise, the breadth of any child node should not exceed that of the root expansion.

To further examine the implications of the above principles and to explore how to prune the draft tree effectively, we conducted analogy experiments with a fixed 9-ary draft tree of height 3. This tree contains $1 + 9 + 9^2 = 91$ nodes, indexed from 0 to 90. Node 0 is the root corresponding to the next-token position; nodes 1-9 form the first layer; and nodes 10-90 form the second layer. Under this level-order indexing scheme, each node $i > 0$ has its parent given by $\text{parent}(i) = \lfloor \frac{i-1}{k} \rfloor$.

As illustrated in Figure 2, the second-layer children of node 1 exhibit an accumulative trend resembling that of the first layer, albeit slightly slower. For children of parents with larger IDs, the growth further slows and the total volume is nearly halved. The distribution remains front-loaded, with the 50th percentile at the second node (ID 11) and the 85th percentile at the 19th node (ID 28, the second child of node 3). Compared with *copy-logit*, the MAT raises from 1.87 to 2.34.

To capture this behavior, we define the breadth allocation in the Logits Tree for child nodes as

$$b_{\text{child}(i,j)} = \max\left(1, \left\lfloor \frac{b_i}{2^{j+[i\neq 0]}} \right\rfloor\right), \quad j = 0, 1, \ldots, b_i - 1. \tag{3}$$

where $b_i$ is the breadth of the parent node and $j$ is the child index. Specifically, nodes at the first layer start with the maximum breadth, while deeper layers inherit half of their parent's breadth. This design ensures that the upper part of the Logits Tree expands more aggressively, while deeper layers are progressively pruned. Given a specific draft capacity, the Logits Tree then expands in a breadth-first manner according to this allocation rule. Figure 6 in the Appendix illustrates this process using a 4-ary example, showing: (i) the original $k$-ary indexing, and (ii) the pruned Logits Tree structure.

## 3.2 RETRIEVAL TREE WITH LRU EVICTION

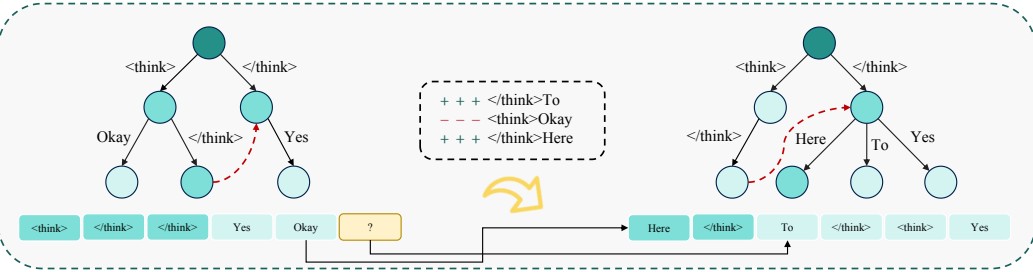

Figure 3: Illustration of the LRU-based eviction strategy in RACER's retrieval automaton. Solid black edges denote standard trie transitions, and dashed red edges denote failure links. Yellow nodes represent unallocated states. Green nodes indicate allocated states, where darker color corresponds to more recent usage. The example demonstrates how inserting the 2-gram [</think>, To] creates a new node To, and how LRU leaf node Okay is evicted and replaced with Here when the capacity is reached with the new 2-gram [</think>, Here].

Approaches such as SAM Decoding (Hu et al., 2025) and LogitSpec (Liu et al., 2025b) indicate that explicit retrieval drafts can complement the logits and improve acceptance rates. Motivated by these findings, we aim to design an efficient retrieval structure that exploits repeated patterns in the context. Classical indexing structures such as suffix arrays (Manber & Myers, 1993) or suffix automata (Blumer et al., 1984) provide efficient substring matching, but they grow proportionally with the context length and lack a natural mechanism to discard obsolete states. This is undesirable in language modeling, where the distribution of substrings follows a Zipf-like long-tail law, implying that many substrings have little utility and can be safely evicted. To balance efficiency and adaptivity, we propose to use an Aho–Corasick (AC) automaton (Aho & Corasick, 1975) to maintain an $n$-gram based Retrieval Tree. Unlike a plain $n$-gram trie, the AC automaton supports failure links that facilitate fast state transitions and naturally enrich draft diversity.

**Transition Rule**  As shown in Figure 3, we incorporate an **LRU-style (Least Recently Used) eviction mechanism** into the AC automaton so that infrequent $n$-gram patterns are pruned while new ones from the incoming context are continually incorporated. Each time a state is visited – either through a valid transition or via a failure-link fallback – that state is marked as "touched". Importantly, when backtraking with failure links, all of its ancestor (prefix) states are also necessarily touched. This ensures that prefix nodes always remain equal or more recent than their descendants. For example, when the token `Yes` follows `[<think>, </think>]` and there is no direct transition, the automaton backtracks along its failure link to state `[</think>]` and then transitions to `[</think>, Yes]`; all states along this fallback and transition path are touched.

**Update and Eviction Rule**  During decoding, newly observed $n$-grams are incrementally inserted into the automaton. Insertion follows the transition path whenever possible; if a transition does not exist, a new node is allocated either from an empty slot or by reusing an LRU leaf node. When the automaton reaches its predefined capacity, the LRU *leaf* node is evicted (e.g., the leaf `Okay` in Figure 3). All nodes are managed using a hash table and a doubly linked list, enabling $\mathcal{O}(1)$ updates and eviction. Failure links are updated lazily: they are rebuilt only once at the end of the prefilling phase. Before the rebuild, the newly added portion of the structure temporarily behaves as a standard trie without failure links.

**Expansion Rule**  We consider all match states (borders) whose matched depth is at least 2. For each such border, we take the sub-trie rooted at that state, collect all outgoing $n$-gram continuations, pool them across borders, and select the globally most frequent top-$k$ continuation states to expand the Retrieval Tree. For illustration, suppose the current match state is `[<think>, </think>]`, and the automaton has observed:

- `[<think>, </think>, Okay]` with frequency 3,
- `[</think>, Yes, <space>]` with frequency 2,
- `[</think>, Yes, <comma>]` with frequency 1.

Pooling continuations over all prefixes ending in `</think>` yields:

$$[\texttt{Okay}]:3, \quad [\texttt{Yes}]:2+1=3, \quad [\texttt{Yes, <space>}]:2, \quad [\texttt{Yes, <comma>}]:1.$$

Selecting the top-3 continuations gives:

$$[\texttt{Okay}]:3, \quad [\texttt{Yes}]:2+1=3, \quad [\texttt{Yes, <space>}]:2,$$

If a depth constraint is applied (i.e., only continuations consistent with the matched suffix of length 2 are allowed), then only the continuation `Okay` remains valid, because it is the only 3-gram whose prefix exactly matches the current border `[<think>, </think>]`. Other candidates such as `[Yes]` or `[Yes, <space>]` originate from shorter matches ending in `</think>`, and are therefore pruned under the depth restriction.

### 3.3 Integrated Strategy

Given a fixed speculative capacity $C$, retrieval-based candidates are first generated according to the expansion rule in Section 3.2, while the remaining capacity is allocated to the Logits Tree via breadth-first expansion (Equation 3). Since retrieval candidates are sparse but structurally reliable, we retain only the most confident ones and leave the remaining budget for logits-based exploration through the top-$k$ adjacency matrix.

Importantly, retrieval not only complements the current speculation, but also provides stronger cues for upcoming tokens than logits alone. Because retrieval captures repeated patterns in closer contexts, it guides the logits distribution toward sharper predictions and mitigates error accumulation in speculative expansions.

The two candidate sets are finally merged into a unified draft tree through a trie-based union, and verified by the target model under the guess-and-verify scheme. This hybrid design enables RACER to exploit both *seen information* from retrieval and *unseen speculation* from logits, achieving higher acceptance rates with controlled memory usage. The overall workflow is presented in Figure 4; further implementation details can be found in Section D.

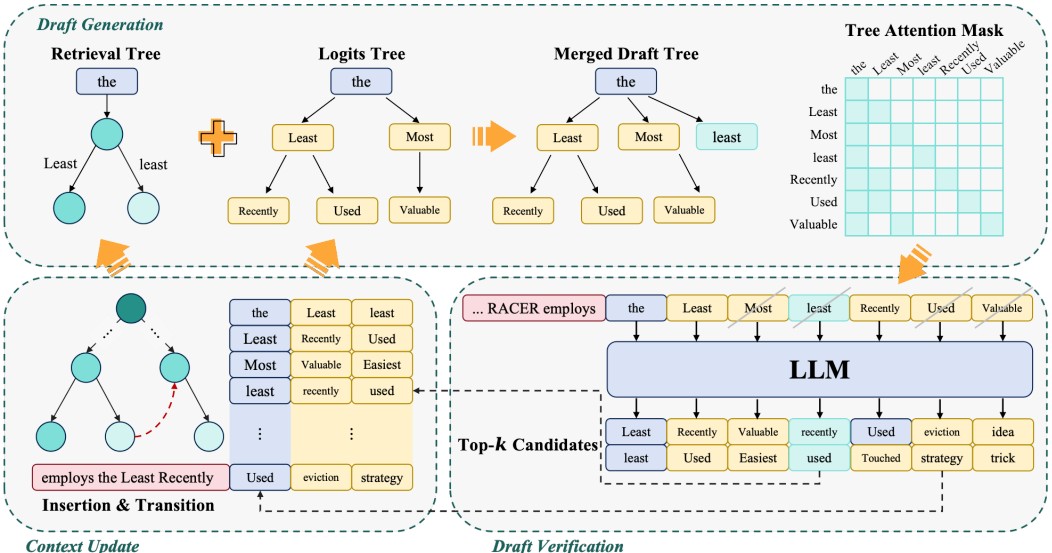

Figure 4: Overview of RACER. At each decoding step, the AC automaton accepts the next token and identifies border nodes with depth $\geq 2$, from which the globally most frequent children are selected as retrieval candidates. If retrieval nodes do not fill the draft capacity $C$, the remaining slots are assigned to Logits Tree expansion (Eq. 3). Verified $n$-grams are inserted into the automaton, while the logits adjacency matrix is refreshed with the newly generated logits.

# 4 EXPERIMENTS

## 4.1 EXPERIMENTAL SETUP

Following prior work (Luo et al., 2025), we focus on greedy decoding with batch size 1 and maximum output length 1024. We report the following metrics: **Mean Accepted Tokens (MAT)** (Xia et al., 2024): the average number of tokens confirmed in a single speculative decoding step; **Throughput (Tokens/s)**: the number of tokens processed per second during inference; **Speedup Ratio**: relative performance compared with HuggingFace's implementation of autoregressive decoding.

For all experiments, we use the following default hyperparameters unless otherwise specified. For the Logits Tree, the maximum breadth is set to 8. For the Retrieval Tree, we maintain up to 10,000 nodes with an $n$-gram length of 10. The draft size of each decoding step is 64 as suggested in Medusa (Cai et al., 2024). These values are chosen based on preliminary analyses, and we further demonstrate in Section 4.3 that our method is robust with respect to these hyperparameters.

Experiments were conducted on two types of GPUs: NVIDIA A800 (80GB) and RTX 4090 (24GB). For practicality, we ran models with 7B/8B parameters on the RTX 4090, and larger models (13B and above) on the A800, so that all models fit comfortably within the available memory while ensuring fair comparison. Detailed hardware and software specifications are included in Appendix A.

**Datasets and Target Models** We evaluate on three benchmarks: **Spec-Bench** (Xia et al., 2024), **HumanEval** (Chen et al., 2021), and **MGSM-ZH** (Cobbe et al., 2021; Shi et al., 2022). Spec-Bench covers diverse scenarios including Multi-turn Conversation (MT), Translation (Trans), Summarization (Sum), Question Answering (QA), Mathematical Reasoning (Math), and Retrieval-Augmented Generation (RAG). HumanEval is a widely used benchmark for code generation. MGSM-ZH is the Chinese counterpart of GSM8K. Following the multilingual evaluation setup used in PEARL (Liu et al., 2025a), we adopt MGSM-ZH to assess non-English mathematical reasoning in a language where Qwen3 exhibits stable performance, while avoiding languages for which Vicuna generates unstable outputs. Together, these benchmarks cover general-purpose, domain-specific, and cross-lingual reasoning tasks. We utilize Vicuna (Chiang et al., 2023) at three scales: 7B, 13B, and 33B,

where the 7B and 13B models use version 1.5 and the 33B model uses version 1.3. We also include Qwen3 (Yang et al., 2025) at the corresponding 8B, 14B, and 32B scales.

**Baselines** We compare RACER against two retrieval-based methods (PLD and REST), two logits-involved methods (Token Recycling and LogitSpec), and the state-of-the-art model-based method EAGLE-3. **PLD** (Saxena, 2023) stores past $n$-grams as sequential keys and their succeeding $m$-grams as predicted values. **REST** (He et al., 2023) builds a suffix array over the training set to locate the longest suffix match, then expands the matched continuation into a trie. **Token Recycling (TR)** (Luo et al., 2025) maintains a top-$k$ adjacency matrix from token logits and extends it into a draft tree using a predefined template. **LogitSpec** (Liu et al., 2025b) speculates the first draft token from the top-$k$ candidates of the last-step logits, then augments expansion with retrieval tokens drawn from the context. **EAGLE-3** (Li et al., 2025) incorporates low-, mid-, and high-level features of the target model, with a Transformer decoder layer as its core. Since the official Qwen3 draft model weights for EAGLE-3 have not been released, we use the re-implementation by AngelSlim[1]. All baselines are run with their default hyperparameters.

## 4.2 MAIN RESULTS

Table 1 reports the performance of RACER compared to baseline methods on different datasets. Among retrieval-based methods, PLD relies solely on the context and REST leverages an external training set. Their speedup ratios remain below $2\times$, highlighting the inherent limitations of retrieval-only approaches. In contrast, methods involving logits – whether model-free or model-based – can readily surpass $2\times$ speedup, demonstrating the advantage of exploiting predictive distributions from the target model.

RACER achieves the best speedup across most benchmarks and consistently delivers the highest overall speedup among different target models. Notably, with Qwen3-series target models, EAGLE-3 (AngelSlim's re-implementation) achieves the highest MAT on most tasks except the Chinese reasoning benchmark MGSM-ZH. However, its advantage in MAT does not translate into end-to-end efficiency, as RACER still outperforms it in terms of speedup ratio. This is because EAGLE-3 requires an additional draft model, incurring extra inference cost, whereas RACER remains lightweight and model-free.

Moreover, the weaker performance of EAGLE-3 on MGSM-ZH highlights a broader limitation of model-based approaches: their effectiveness is sensitive to the distribution and coverage of the draft model's training data. It is plausible that AngelSlim's re-implementation, trained primarily on English corpora, fails to simulate the target model's distribution accurately in Chinese reasoning tasks. Such data distribution mismatches, rooted in differences in post-training procedures or training corpora, generally constrain the robustness of model-based speculative decoding methods. To complement this observation, we additionally report results on purely English reasoning benchmarks GSM8K (Cobbe et al., 2021), AIME (Veeraboina, 2023) and MATH (Hendrycks et al., 2021) in Appendix F, along with extended discussions on how EAGLE-3 compares to RACER.

Compared with the other two logits-involved methods, TR and LogitSpec, RACER consistently outperforms them on both MAT and speedup. Overall, TR achieves better performance than LogitSpec, except on the reasoning task MGSM-ZH. This suggests that in general tasks, TR is able to exploit logits more effectively. However, in reasoning tasks where repeated patterns from previous context are frequent, retrieval provides crucial guidance and brings substantial benefits. Therefore, an effective strategy must integrate both logits and retrieval. RACER achieves this integration successfully, yielding consistently superior MAT and speedup across benchmarks.

In summary, RACER consistently delivers the best trade-off between acceptance and efficiency, achieving stable improvements across model sizes, domains, and languages, thus demonstrating its robustness and generality compared to retrieval-only, logits-only, and model-based baselines.

## 4.3 ABLATION STUDIES

To better verify RACER, we conduct ablation experiments with Vicuna-7B-v1.5 on Spec-Bench, HumanEval and MGSM-ZH.

---

[1]https://github.com/Tencent/AngelSlim

Table 1: Results on Spec-Bench, HumanEval, and MGSM-ZH, reported in mean accepted tokens (MAT) and speedup ratio. Best results are in **bold**, suboptimal results are underlined. PLD and REST belong to retrieval-based methods, whereas TR is solely logits-based, LogitSpec integrates both retrieval and logits, and EAGLE-3 represents a **model-based** approach.

| Models | Method | Spec-Bench | | HumanEval | | MGSM-ZH | | Average | |
|--------|--------|------|---------|------|---------|------|---------|------|---------|
| | | MAT | Speedup | MAT | Speedup | MAT | Speedup | MAT | Speedup |
| Vicuna 7B | PLD | 1.71 | 1.50 | 1.58 | 1.40 | 2.57 | 2.27 | 1.95 | 1.87 |
| | REST | 1.82 | 1.45 | 2.06 | 1.71 | 1.29 | 1.06 | 1.72 | 1.41 |
| | LogitSpec | 2.34 | 1.77 | 2.22 | 1.66 | 3.55 | 2.67 | 2.70 | 2.03 |
| | TR | 2.76 | 2.06 | 2.83 | 2.17 | 3.00 | 2.30 | 2.86 | 2.18 |
| | **RACER** | **3.00** | **2.21** | **3.11** | **2.29** | **3.71** | **2.77** | **3.27** | **2.42** |
| Vicuna 13B | PLD | 1.65 | 1.41 | 1.59 | 1.43 | 2.45 | 2.11 | 1.90 | 1.65 |
| | REST | 1.82 | 1.44 | 2.07 | 1.71 | 1.31 | 1.07 | 1.73 | 1.41 |
| | LogitSpec | 2.32 | 1.73 | 2.23 | 1.77 | 3.44 | 2.72 | 2.66 | 2.07 |
| | TR | 2.79 | 1.99 | 2.83 | 2.08 | 3.05 | 2.22 | 2.89 | 2.10 |
| | **RACER** | **2.95** | **2.25** | **3.09** | **2.42** | **3.64** | **2.83** | **3.23** | **2.50** |
| Vicuna 33B | PLD | 1.33 | 1.03 | 1.64 | 1.48 | 2.18 | 1.97 | 1.72 | 1.49 |
| | REST | 1.81 | 1.54 | 1.98 | 1.72 | 1.32 | 1.17 | 1.70 | 1.48 |
| | LogitSpec | 2.32 | 1.73 | 2.35 | 1.92 | 2.96 | 2.44 | 2.54 | 2.03 |
| | TR | 2.63 | 1.83 | 2.79 | 2.05 | 2.83 | 2.10 | 2.75 | 1.99 |
| | **RACER** | **2.74** | **2.20** | **3.16** | **2.58** | **3.36** | **2.77** | **3.09** | **2.52** |
| Qwen3 8B | PLD | 1.52 | 1.35 | 1.52 | 1.41 | 1.69 | 1.52 | 1.58 | 1.43 |
| | EAGLE-3[†] | **3.46** | **2.14** | **3.84** | **2.44** | 1.41 | 0.86 | **2.90** | 1.81 |
| | **RACER** | 2.73 | 2.13 | 2.79 | 2.24 | **2.95** | **2.26** | 2.82 | **2.21** |
| Qwen3 14B | PLD | 1.45 | 1.34 | 1.43 | 1.27 | 1.59 | 1.49 | 1.49 | 1.37 |
| | EAGLE-3[†] | **2.72** | 1.87 | **3.03** | 2.05 | 1.56 | 1.12 | 2.44 | 1.68 |
| | **RACER** | 2.67 | **2.23** | 2.77 | **2.29** | **2.88** | **2.30** | **2.77** | **2.27** |
| Qwen3 32B | PLD | 1.45 | 1.34 | 1.34 | 1.23 | 1.56 | 1.46 | 1.45 | 1.34 |
| | EAGLE-3[†] | **2.88** | 2.12 | **2.97** | **2.17** | 1.60 | 1.18 | 2.48 | 1.82 |
| | **RACER** | 2.66 | **2.17** | 2.55 | 2.08 | **2.78** | **2.28** | **2.66** | **2.18** |

[†] EAGLE-3 models and weights from AngelSlim's re-implementation.

Table 2: Ablation experiments on Spec-Bench, HumanEval and MGSM-ZH.

| Models | Method | Spec-Bench | | HumanEval | | MGSM-ZH | |
|--------|--------|------|---------|------|---------|------|---------|
| | | MAT | Speedup | MAT | Speedup | MAT | Speedup |
| Vicuna 7B | w/o logits | 1.59↓1.41 | 1.43↓0.78 | 1.67↓0.73 | 1.52↓0.77 | 2.39↓1.32 | 2.11↓0.66 |
| | w/o retrieval | 2.72↓0.28 | 2.01↓0.20 | 2.68↓0.20 | 2.04↓0.25 | 2.95↓0.76 | 2.23↓0.54 |
| | **RACER** | **3.00** | **2.21** | **3.11** | **2.29** | **3.71** | **2.77** |
| Vicuna 13B | w/o logits | 1.56↓1.39 | 1.38↓0.87 | 1.65↓1.44 | 1.43↓0.99 | 2.29↓1.35 | 1.95↓0.88 |
| | w/o retrieval | 2.68↓0.27 | 2.06↓0.19 | 2.70↓0.39 | 2.14↓0.28 | 2.98↓0.66 | 2.34↓0.59 |
| | **RACER** | **2.95** | **2.25** | **3.09** | **2.42** | **3.64** | **2.83** |
| Vicuna 33B | w/o logits | 1.46↓1.28 | 1.38↓0.82 | 1.76↓1.40 | 1.66↓0.92 | 2.10↓1.26 | 1.97↓0.80 |
| | w/o retrieval | 2.55↓0.19 | 2.05↓0.15 | 2.66↓0.50 | 2.19↓0.39 | 2.74↓0.62 | 2.27↓0.50 |
| | **RACER** | **2.74** | **2.20** | **3.16** | **2.58** | **3.36** | **2.77** |

**Component Contribution**   Table 2 reports ablation results by removing either the logits or retrieval component. We observe that removing logits causes the most severe degradation: MAT drops by more than one token on average and speedup decreases by 0.8-1.0×, confirming that logits form the backbone of speculative expansion. In contrast, removing retrieval leads to smaller but still notable drops, especially on MGSM-ZH where MAT and speedup decrease by up to 0.7 and 0.6, respectively. This highlights the complementary role of retrieval in reasoning and cross-lingual tasks, where repeated patterns provide strong predictive cues. Across all three model scales, RACER consistently benefits from both components, validating our integration strategy that balances generalization from logits with structural guidance from retrieval. For more verification of retrieval component, please see Appendix D.4.

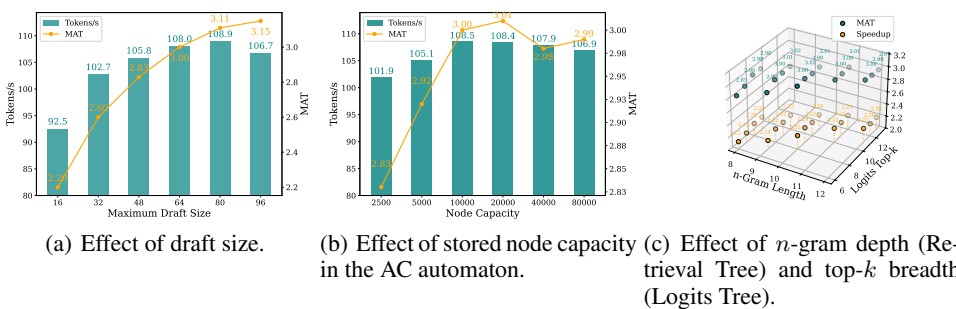

(a) Effect of draft size.    (b) Effect of stored node capacity in the AC automaton.    (c) Effect of $n$-gram depth (Retrieval Tree) and top-$k$ breadth (Logits Tree).

Figure 5: Ablation studies of RACER on key parameters.

**Parameter Robustness**   We further study the robustness of RACER under different hyperparameter settings on Vicuna-7B-v1.5. Conceptually, these hyperparameters control complementary aspects of the unified speculative draft: the *draft size* specifies the total number of draft tokens per step, the *top-$k$ breadth* of the Logits Tree controls how widely we explore model-predicted candidates, and the *capacity* and *$n$-gram depth* of the Retrieval Tree determine how many $n$-grams can be stored and how long they can be. A larger draft size or broader trees generally increases coverage and acceptance probability, but if they grow too large, the decoding regime may shift from memory-bound to compute-bound, leading to diminishing or even negative returns. Similarly, increasing retrieval capacity and $n$-gram depth allows matching more rare patterns, but over-emphasizing long-tail matches can dilute the draft budget and reduce MAT. In practice, RACER remains stable over a broad range of these settings.

Figure 5 summarizes three ablation experiments. **Draft Size (Figure a).** The draft size controls the *total* number of speculative tokens proposed by both the Logits Tree and the Retrieval Tree in each step. Increasing the draft size from 16 to 64 steadily improves both MAT and throughput, after which the gains saturate. This shows that RACER benefits from moderately larger drafts while remaining stable even with further expansion. However, overly large drafts can push the system into a compute-bound regime, where the additional verification cost outweighs the benefit of higher acceptance. The optimal draft size is therefore hardware-dependent: on resource-constrained or edge devices, it is often preferable to cap the draft size to match the device's most efficient batch inference mode. **Retrieval Node Capacity (Figure b).** The Retrieval Tree is implemented as an AC automaton whose *capacity* specifies an upper bound on the number of $n$-gram states (e.g., 10k nodes). Expanding the automaton's storage from 2.5k to around 10k-20k nodes yields the best trade-off between MAT and throughput. Beyond this range, performance only fluctuates slightly, suggesting that RACER does not rely on excessively large retrieval buffers. The built-in LRU eviction policy exploits both *temporal* and *spatial* locality: frequently reused $n$-grams are retained, while rarely used ones are pruned. Since each node in an AC automaton has a unique parent and a failure link, the space complexity of the automaton $\mathcal{A}$ is $\mathcal{O}(|\mathcal{A}|)$, i.e., linear in the number of nodes. In practice, this overhead is modest and remains negligible even with node sizes up to 10k. **Joint Effect of $n$-gram Depth and Top-$k$ Breadth (Figure c).** The $n$-gram depth controls the maximum height of the Retrieval Tree, and thus the longest context it can match. With an upper bound such as $n = 10$, the automaton can flexibly match keys of length 1 to 9 and retrieve the corresponding value sequences. Because candidates are selected according to empirical frequency, this design naturally balances ex-

ploration and exploitation: it covers diverse patterns while prioritizing high-yield matches. On the Logits Tree side, the top-$k$ breadth specifies how many high-probability continuations are expanded at each level; increasing it extends coverage into the long tail but also competes for the finite draft budget. The 3D plot shows that both MAT and speedup improve smoothly as $n$-gram depth and top-$k$ breadth increase, and the performance surface remains relatively flat near the optimal range ($n$-gram depth 9-11, top-$k$ breadth 8-10), indicating that RACER is robust to small parameter deviations. This trend is consistent with our analysis in Section 3.1, where the 85th percentile rank for the *copy-logit* strategy is 9, suggesting that setting these parameters around this range is sufficient and provides a practical guideline when adapting RACER to new target models.

## 5 RELATED WORK

Efficient inference is crucial for real-time applications and resource-constrained scenarios. Various strategies, including KV cache compression and quantization, have been developed to reduce latency. Among these, speculative decoding (SD) (Leviathan et al., 2023; Chen et al., 2023) stands out as a promising technique that predicts multiple continuations simultaneously, reducing decoding steps while maintaining accuracy.

Speculative decoding methods can be broadly categorized into **draft-model-based** and **draft-model-free** approaches. Draft-model-based methods use additional models to predict draft tokens. These models are typically (i) separately trained or (ii) derived from smaller variants of the same model family. Some methods reuse hidden states to predict multiple future tokens in parallel (Cai et al., 2024; Li et al., 2024b;a), employing different layer selection and draft tree expansion strategies. Beyond post-training draft models, Multi-Token Prediction (MTP) integrates draft generation during pre-training. Gloeckle et al. (2024) proposes parallel prediction of multiple tokens with independent output heads, while Liu et al. (2024) introduces sequential multi-token prediction to preserve the full causal chain at each prediction depth.

In contrast, **draft-model-free** methods, such as retrieval-based approaches, eliminate the need for additional models by constructing retrieval libraries to generate draft tokens. PLD (Saxena, 2023) builds libraries from past content, achieving speedup in tasks with high redundancy like summarization. However, these methods are limited by their inability to generate novel content or adapt to diverse queries. REST (He et al., 2023) builds retrieval libraries from existing corpora, offering substantial speedup but facing challenges such as large memory requirements and retrieval inefficiencies. Token Recycling (Luo et al., 2025) requires no additional generation, covering a broader range of continuations using past logits, while minimizing storage and retrieval costs.

RACER integrates both retrieval-based and logits-based methods, unifying the strengths of both. By combining logits for speculative prediction and retrieval for structural guidance, RACER achieves superior speedup and acceptance rates across benchmarks, while maintaining lightweight memory usage suitable for resource-constrained scenarios.

## 6 CONLUSION

In this work, we introduced RACER, a training-free framework that unifies retrieval-based and logits-based signals for speculative decoding. By treating retrieved exact patterns as structural anchors and logits as dynamic future cues, RACER constructs richer speculative drafts while remaining lightweight and plug-and-play. Extensive experiments across multiple model families (Vicuna, Qwen3), hardware settings (RTX 4090, A800), and benchmarks (Spec-Bench, HumanEval, MGSM-ZH) demonstrate consistent acceleration, stable memory usage, and improved speculative efficiency measured by MAT, tokens-per-second, and speedup ratio. Our method remains robust under diverse hyperparameter choices, underscoring both practicality and scalability. We believe RACER establishes a general foundation for training-free speculative decoding, opening avenues for future work on integrating more advanced retrieval structures, multilingual retrieval cues, and harmonization with parallel or distributed decoding algorithms.

## ETHICS STATEMENT

This work focuses on improving the efficiency of LLM inference through training-free speculative decoding in real-time and resource-constrained environments. We do not introduce or rely on additional training data, and our experiments are conducted entirely on publicly available benchmarks (Spec-Bench, HumanEval, and MGSM-ZH) and widely used open-source models (Vicuna, Qwen3). As such, no sensitive, private, or personally identifiable information is involved in this research.

## REPRODUCIBILITY STATEMENT

We have made efforts to ensure the reproducibility of all reported results. Detailed descriptions of hardware (RTX 4090 and A800 GPUs), software (PyTorch, CUDA, HuggingFace Transformers versions), and experimental setups are provided in Appendix A. All hyperparameters, including maximum draft size, retrieval node capacity, $n$-gram depth, and top-$k$ breadth, are explicitly specified in Section 4. Baselines are reproduced with their official implementations or widely accepted re-implementations (e.g., AngelSlim's EAGLE-3). To further facilitate verification, we provide pseudocode for key algorithmic components such as Logits Tree construction and the LRU eviction strategy in Appendix D. Our implementation (including evaluation scripts and inference CLI) is released in this anonymous repository, and will be made publicly available upon publication.

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

## A EXPERIMENTAL SETUP

**Hardware Setup**  Experiments were conducted on dual-GPU servers, with all runs restricted to a single GPU to ensure fairness and reproducibility. We used an NVIDIA RTX 4090 (24GB) with 20 CPU cores for 7B/8B-scale models, and an NVIDIA A800 (80GB) with 64 CPU cores for 13B-scale and larger models.

**Software Setup**  Our implementation is based on PyTorch and HuggingFace Transformers. Experiments were run under the following environment:

- PyTorch `2.8.0` with CUDA `12.8` and cuDNN `91002`
- HuggingFace Transformers `4.37.1` for Vicuna experiments, and `4.52.3` for Qwen3 experiments

We enabled `fp16` inference on both GPUs. No further optimizations (e.g., quantization or TensorRT) were applied, to ensure fair comparison with prior work.

**Evaluation Instructions**  In our experiments, we employ different instructions for different evaluation tasks and models. For Vicuna, we use its standard instructions:

---

**Chat Template for Vicuna on Spec-Bench and MGSM**

A chat between a curious user and an artificial intelligence assistant. The assistant gives helpful, detailed, and polite answers to the user's questions.

**USER:** Question

**ASSISTANT:**

---

---

**Chat Template for Vicuna on HumanEval**

A chat between a curious user and an artificial intelligence assistant. The assistant gives helpful, detailed, and polite answers to the user's questions.

**USER:** Implement the following code. `Code`

**ASSISTANT:**

---

For Qwen3, we use a common system prompt:

---

**Chat Template for Qwen3 on Spec-Bench and MGSM**

You are a helpful assistant.

**USER:** `Question`

**ASSISTANT:**

---

**Chat Template for Qwen3 on HumanEval**

You are a helpful assistant.

**USER:** Implement the following code. `Code`

**ASSISTANT:**

---

## B  ADDITIONAL EXPERIMENT RESULTS

Table 3 and Table 4 present the results on individual SpecBench tasks, complementing the overall comparison in Table 1. RACER consistently outperforms other model-free methods in most cases, including all overall speedup ratios, with only a few exceptions in Translation (Trans), Question Answering (QA), and Mathematical Reasoning (Math). For Translation, the retrieval component contributes little to MAT and may even offset part of the logits-based advantage, leading to weaker performance on smaller models. However, with larger models such as Vicuna-33B, this effect becomes negligible, and RACER consistently outperforms TR. For QA, TR surpasses RACER on Vicuna-7B across both A800 and RTX 4090, suggesting that its predefined tree template may align better with the characteristics of this task. For Math, TR slightly outperforms RACER only on Vicuna-7B with A800, but this advantage does not generalize to other model scales or hardware. In contrast, as model size increases, RACER shows a growing margin in overall speedup over TR, highlighting its robustness across diverse tasks and architectures.

Table 5 complements Table 1 by providing a closer look at Vicuna-7B across different hardware. We observe that LogitSpec achieves slightly lower MAT on the A800 compared to the RTX 4090, but exhibits a higher speedup ratio on the A800. This suggests that its efficiency gains are more sensitive to hardware characteristics, whereas RACER maintains stable improvements across both platforms.

## C  ADDITIONAL ABLATION RESULTS

Table 6 reports the ablation results on Spec-Bench across six tasks. The results clearly show that removing either logits or retrieval consistently harms performance, confirming that both components are essential for RACER. **Without logits**: Performance drops significantly across all tasks, often by more than $0.7\times$ in speedup. This highlights that logits are the dominant factor for efficient speculation. **Without retrieval**: The degradation is generally smaller but still noticeable, especially on tasks such as MT and Sum, where repeated patterns and structural cues play a larger role. **Full RACER**: By integrating both signals, RACER achieves balanced improvements and consistently outperforms the ablated variants, showing robustness across tasks, model scales, and hardware platforms. This

Table 3: Speedup ratios and overall MAT across different tasks of Spec-Bench evaluated on NVIDIA A800 (80GB).

| Models | Method | MT | Trans | Sum | QA | Math | RAG | MAT | Speedup |
|--------|--------|----|-------|-----|----|------|-----|-----|---------|
| Vicuna 7B | PLD | 1.42 | 0.97 | 2.25 | 1.12 | 1.59 | 1.61 | 1.72 | 1.49 |
| | REST | 1.52 | 1.09 | 1.21 | 1.28 | 1.07 | 1.31 | 1.82 | 1.33 |
| | LogitSpec | 1.79 | 1.35 | 2.48 | 1.50 | 2.08 | 1.82 | 2.35 | 1.86 |
| | TR | 2.19 | **1.84** | 2.02 | **2.02** | **2.58** | 1.85 | 2.76 | 2.15 |
| | **RACER** | **2.23** | 1.61 | **2.61** | 1.82 | 2.46 | **2.12** | **3.01** | **2.22** |
| Vicuna 13B | PLD | 1.36 | 0.98 | 1.93 | 1.09 | 1.53 | 1.44 | 1.65 | 1.41 |
| | REST | 1.61 | 1.14 | 1.30 | 1.56 | 1.21 | 1.42 | 1.82 | 1.44 |
| | LogitSpec | 1.67 | 1.33 | 2.13 | 1.40 | 2.00 | 1.72 | 2.32 | 1.73 |
| | TR | 2.00 | **1.74** | 1.89 | 1.82 | 2.31 | 1.87 | 2.79 | 1.99 |
| | **RACER** | **2.23** | 1.70 | **2.49** | **1.85** | **2.55** | **2.21** | **2.95** | **2.25** |
| Vicuna 33B | PLD | 1.33 | 1.03 | 1.84 | 1.11 | 1.57 | 1.23 | 1.54 | 1.37 |
| | REST | 1.70 | 1.24 | 1.41 | 1.57 | 1.31 | 1.61 | 1.81 | 1.54 |
| | LogitSpec | 1.66 | 1.36 | 2.11 | 1.38 | 2.02 | 1.50 | 2.11 | 1.71 |
| | TR | 1.90 | 1.67 | 1.85 | 1.76 | 2.20 | 1.72 | 2.63 | 1.83 |
| | **RACER** | **2.22** | **1.73** | **2.41** | **1.87** | **2.51** | **1.91** | **2.74** | **2.20** |
| Qwen3 8B | PLD | 1.37 | 1.67 | 1.35 | 1.41 | 1.72 | 1.44 | 1.52 | 1.47 |
| | EAGLE-3 | **2.43** | 2.07 | 2.12 | 2.36 | 2.63 | **2.45** | **3.47** | 2.37 |
| | **RACER** | 2.41 | **2.69** | **2.35** | **2.41** | **2.72** | 2.42 | 2.73 | **2.48** |
| Qwen3 14B | PLD | 1.27 | 1.56 | 1.18 | 1.31 | 1.54 | 1.29 | 1.45 | 1.34 |
| | EAGLE-3 | 1.93 | 1.86 | 1.60 | 1.82 | 2.12 | 1.76 | **2.72** | 1.87 |
| | **RACER** | **2.12** | **2.55** | **2.10** | **2.19** | **2.47** | **2.11** | 2.67 | **2.23** |
| Qwen3 32B | PLD | 1.26 | 1.54 | 1.13 | 1.35 | 1.48 | 1.29 | 1.44 | 1.33 |
| | EAGLE-3 | **2.20** | 2.03 | 1.84 | 2.02 | **2.47** | 2.04 | **2.88** | 2.12 |
| | **RACER** | 2.10 | **2.40** | **2.04** | **2.06** | 2.38 | **2.16** | 2.66 | **2.17** |

Table 4: Speedup ratios and overall MAT across different tasks of Spec-Bench evaluated on NVIDIA RTX 4090 (24GB).

| Models | Method | MT | Trans | Sum | QA | Math | RAG | MAT | Speedup |
|--------|--------|----|-------|-----|----|------|-----|-----|---------|
| Vicuna 7B | PLD | 1.43 | 1.00 | 2.19 | 1.16 | 1.63 | 1.56 | 1.71 | 1.50 |
| | REST | 1.63 | 1.18 | 1.30 | 1.53 | 1.21 | 1.46 | 1.82 | 1.45 |
| | LogitSpec | 1.70 | 1.27 | 2.37 | 1.42 | 1.95 | 1.76 | 2.34 | 1.77 |
| | TR | 2.10 | **1.76** | 1.98 | **1.91** | 2.41 | 1.81 | 2.76 | 2.06 |
| | **RACER** | **2.21** | 1.65 | **2.54** | 1.82 | **2.45** | **2.09** | **3.00** | **2.21** |
| Qwen3 8B | PLD | 1.27 | 1.57 | 1.24 | 1.30 | 1.56 | 1.32 | 1.52 | 1.35 |
| | EAGLE-3 | **2.19** | 1.92 | 1.88 | **2.16** | **2.41** | **2.14** | **3.46** | **2.14** |
| | **RACER** | 2.05 | **2.44** | **1.96** | 2.11 | 2.37 | 2.02 | 2.73 | 2.13 |

task-dependent effect aligns with our main results and further validates the complementary nature of logits- and retrieval-based speculation.

Table 5: Experimental results of Vicuna-7B-v1.5 and Qwen3-8B on Spec-Bench, HumanEval, and MGSM-ZH, running on the NVIDIA A800 GPU (80G).

| Models | Method | Spec-Bench | | HumanEval | | MGSM-ZH | | Average | |
|---|---|---|---|---|---|---|---|---|---|
| | | MAT | Speedup | MAT | Speedup | MAT | Speedup | MAT | Speedup |
| | PLD | 1.72 | 1.49 | 1.59 | 1.43 | 2.57 | 2.10 | 1.96 | 1.67 |
| | REST | 1.82 | 1.33 | 2.06 | 1.59 | 1.29 | 0.91 | 1.72 | 1.92 |
| Vicuna 7B | LogitSpec | 2.35 | 1.86 | 2.23 | 1.84 | 3.57 | **2.81** | 2.72 | 2.17 |
| | TR | 2.76 | 2.15 | 2.83 | 2.31 | 3.00 | 2.27 | 2.86 | 2.24 |
| | **RACER** | **3.01** | **2.22** | **3.10** | **2.35** | **3.73** | 2.57 | **3.28** | **2.38** |
| | PLD | 1.52 | 1.47 | 1.51 | 1.41 | 1.68 | 1.69 | 1.57 | 1.52 |
| Qwen3 8B | EAGLE-3 | **3.47** | 2.37 | **3.84** | **2.70** | 1.40 | 1.06 | **2.90** | 2.04 |
| | **RACER** | 2.73 | **2.48** | 2.81 | 2.40 | **2.96** | **2.66** | 2.83 | **2.51** |

Table 6: Ablation experiments on multiple tasks of Spec-Bench (Speedup only).

| Models | Method | MT | Trans | Sum | QA | Math | RAG |
|---|---|---|---|---|---|---|---|
| | w/o logits | 1.42↓0.79 | 0.94↓0.71 | 1.85↓0.69 | 1.11↓1.71 | 1.51↓0.94 | 1.50↓0.59 |
| Vicuna 7B | w/o retrieval | 1.98↓0.23 | 1.68↑0.03 | 2.14↓0.40 | 1.81↓1.02 | 2.31↓0.14 | 1.90↓0.19 |
| | **RACER** | **2.21** | **1.65** | **2.54** | **2.82** | **2.45** | **2.09** |
| | w/o logits | 1.35↓0.88 | 0.90↓0.80 | 1.69↓0.80 | 1.04↓0.71 | 1.51↓1.04 | 1.56↓0.56 |
| Vicuna 13B | w/o retrieval | 2.04↓0.19 | 1.69↓0.01 | 2.17↓0.32 | 1.81↓0.04 | 2.37↓0.18 | 2.00↓0.21 |
| | **RACER** | **2.23** | **1.70** | **2.49** | **1.85** | **2.55** | **2.21** |
| | w/o logits | 1.39↓0.83 | 0.97↓0.76 | 1.63↓0.78 | 1.09↓0.78 | 1.56↓0.79 | 1.26↓0.65 |
| Vicuna 33B | w/o retrieval | 2.03↓0.19 | 1.72↓0.01 | 2.12↓0.29 | 1.84↓0.03 | 2.36↓0.15 | 1.85↓0.06 |
| | **RACER** | **2.22** | **1.73** | **2.41** | **1.87** | **2.51** | **1.91** |

Table 7: Ablation experiments with Qwen2.5 series on NVIDIA A800 (80GB). Overall MAT and speedup ratios on general dataset Spec-Bench are reported.

| | Qwen2.5-0.5B | Qwen2.5-1.5B | Qwen2.5-14B | Qwen2.5-32B |
|---|---|---|---|---|
| MAT | 3.30 | 3.25 | 2.64 | 2.71 |
| Speedup | 2.64 | 2.64 | 2.33 | 2.19 |

Tables 7 and 8 report ablation results across different model sizes and architectures on Spec-Bench under `fp16` on a single NVIDIA A800 (80GB). The Qwen2.5 series are dense instruct models, while the Qwen3 series evaluated here are dense thinking models.

For Qwen2.5, we observe that both MAT and speedup are higher for the smaller models (0.5B and 1.5B), and drop when moving to 14B and 32B. A key factor behind this gap is the output length: on Spec-Bench, Qwen2.5-14B and Qwen2.5-32B generate on average about $1.2\times$ more tokens than the 0.5B and 1.5B models. Since the later tokens rely on longer-range context and are more likely to be rejected, which naturally reduces MAT.

Qwen3 provides a complementary perspective. As thinking models, all Qwen3 variants tend to produce much longer answers than Qwen2.5, so the difference in average output length across Qwen3-0.6B to Qwen3-32B is relatively small. In this regime, the dominant factor for MAT is no longer model size, but long-range dependency itself. This explains why the MAT for Qwen3 stays within a narrow band (2.66 - 2.89): RACER is operating under consistently longer effective sequence lengths,

Table 8: Ablation experiments with Qwen3 series on NVIDIA A800 (80GB). Overall MAT and speedup ratios on general dataset Spec-Bench are reported.

|  | Qwen3-0.6B | Qwen3-1.7B | Qwen3-4B | Qwen3-8B | Qwen3-14B | Qwen3-32B |
|---|---|---|---|---|---|---|
| MAT | 2.89 | 2.78 | 2.78 | 2.73 | 2.67 | 2.66 |
| Speedup | 2.55 | 2.48 | 2.53 | 2.13 | 2.23 | 2.17 |

and its logits-based component is increasingly challenged by dependencies far from the current position due to Transformer nature. Nonetheless, even under these longer-context conditions, RACER is still able to maintain MAT close to 3 on average.

Given that MAT remains roughly stable within each model family, the slight degradation in speedup as model size increases is expected and does not undermine RACER's core advantage. Larger models incur higher verification cost per token, so the same MAT translates into a smaller relative speedup – a general phenomenon shared by speculative decoding methods. The key takeaway from these ablations is that RACER sustains strong and stable acceleration (consistently above $2\times$) across a wide range of model sizes and sequence lengths, even as longer outputs and long-range dependencies make speculation inherently more difficult.

## D   MORE IMPLEMENTATION DETAILS

### D.1   LOGITS TREE CONSTRUCTION

The Logits Tree leverages the top-$k$ adjacency matrix from past logits and expands breadth-first, following the breadth allocation rule in Eq. 3. The full procedure is described in Algorithm 1. Figure 6 illustrates how Logits Tree is pruned from dense to sparse, with the same capacity of 21.

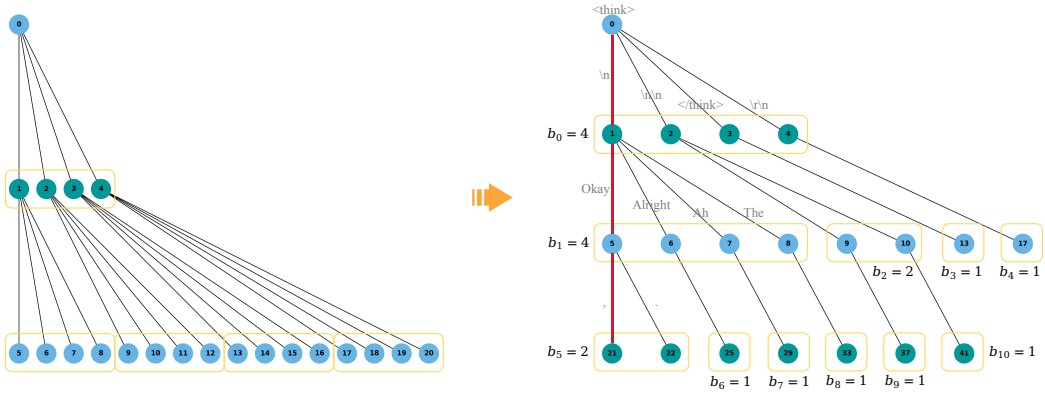

Figure 6: The tree on the left shows the expansion of an unpruned 4-ary tree with 21 nodes, while the tree on the right depicts the expansion of the pruned 4-ary tree with the same number of nodes. The path in red represent a possible candidate [<think>, <end_of_line>, Okay, <comma>].

### D.2   AHO–CORASICK AUTOMATON CONSTRUCTION AND TRANSITION

The Aho–Corasick automton could be simply described as trie with failure links. Each failure link at a node connects to the longest proper suffix of the string at that node, which also serves as a prefix for another pattern in the trie. If no such suffix exists, the link reverts to the root. This is analogous to the "failure function" in the Knuth–Morris–Pratt (KMP) string-matching algorithm (Knuth et al., 1977), but Aho–Corasick extends this idea to work efficiently for multiple patterns.

Figure 8 illustrates the AC automaton's structure, showcasing failure links in red and final states with double circles, though some transitions may be omitted for clarity. The process begins with

**Algorithm 1** Construct Logits Tree (BFS) and Return Draft Candidates

1: **function** BUILDLOGITSTREE(next_token, $C$)      ▷ $C$ is the maximum number of draft nodes
2:      Initialize an empty queue $Q$
3:      Initialize an empty list candidates
4:      token(0) ← next_token
5:      **push** $Q \leftarrow 0$
6:      $C \leftarrow C - 1$      ▷ Root consumes one draft slot
7:      **while** not $Q$.isEmpty() **do**
8:          $u \leftarrow Q$.dequeue()
9:          **if** $C > 0$ **then**
10:              next_breadth $\leftarrow \begin{cases} b(\mathbf{u}), & \text{if } u = \text{root} \\ \lfloor b(\mathbf{u})/2 \rfloor, & \text{otherwise} \end{cases}$      ▷ Breadth allocation follows Eq. 3
11:              **for** $j \leftarrow 0$ **to** $b(u)$ - 1 **do**
12:                  **if** $C = 0$ **then**
13:                      **break**
14:                  **end if**
15:                  $v \leftarrow k \times u + j$      ▷ The $j$-th child of $u$
16:                  token($v$) ← top_k[token($u$)][j]
17:                  $b(v) \leftarrow$ next_breadth
18:                  **push** $Q \leftarrow v$
19:                  next_breadth $\leftarrow \lfloor$ next_breadth$/2 \rfloor$      ▷ Breadth allocation follows Eq. 3
20:                  $C \leftarrow C - 1$
21:              **end for**
22:          **else**      ▷ Backtrack to get a draft candidate
23:              path ← [ ];   $v \leftarrow u$
24:              **while** $v \neq \epsilon$ **do**
25:                  **append** token($v$) to path
26:                  $v \leftarrow$ parent($v$)
27:              **end while**
28:              **reverse**(path)      ▷ Leaf→root to root→leaf
29:              **append** path to candidates
30:          **end if**
31:      **end while**
32:      **return** candidates
33: **end function**

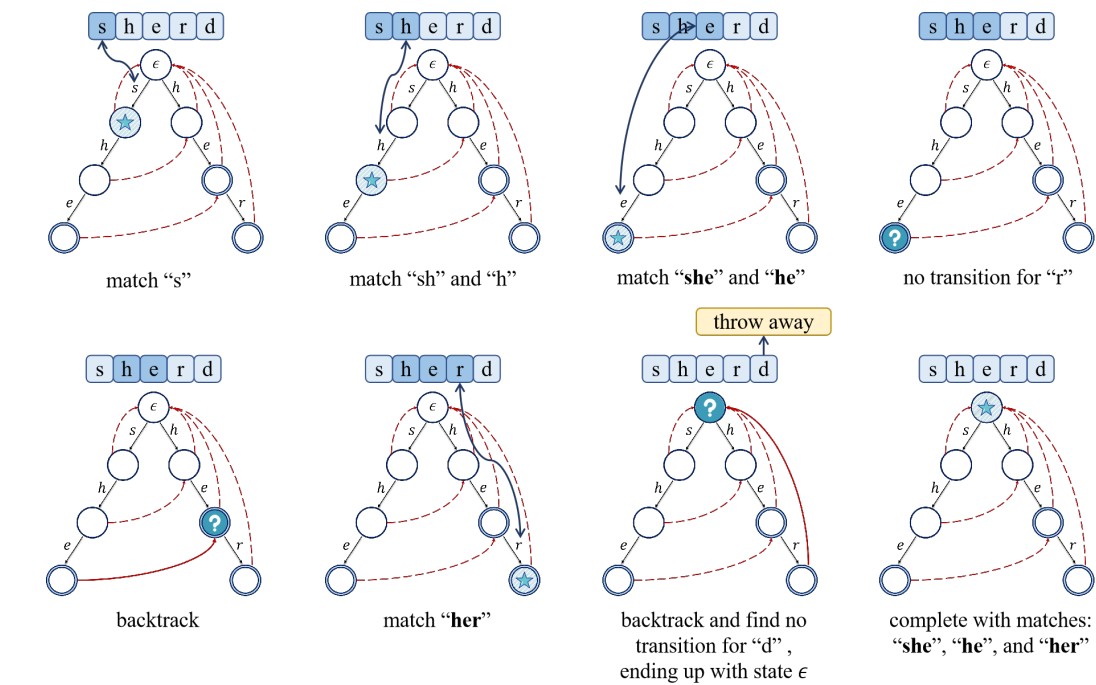

match "s"  match "sh" and "h"  match "**she**" and "**he**"  no transition for "r"

backtrack  match "**her**"  backtrack and find no transition for "d", ending up with state $\epsilon$  complete with matches: "**she**", "**he**", and "**her**"

Figure 7: The process of how "sherd" matches patterns "she", "he", and "her" by transitions on an AC automaton.

an input sequence that progresses through the trie. If a mismatch occurs, such as when at the state "she" and the next input is "r" without a corresponding edge, the automaton utilizes failure links to backtrack until a valid node with the "r" edge is found or until it returns to the root. When the automaton reaches the state "her", not only is the pattern "her" itself recognized but the state of its failure pointer is also included. This forms part of a recursive process: matching a state involves sequentially matching the state of its failure pointer until it traces back to the root node, which represents the absence of further matches, denoted as $\epsilon$.

For further clarification, Figure 7 illustrates the matching process for the string "sherd", identifying the substrings "she", "he", and "her". The elements highlighted in dark blue represent both the longest pattern prefix that the current state can match, and the minimal suffix information necessary for subsequent matches. This setup can be visualized as a "sliding window" that moves from left to right across each position. During normal transitions, this sliding window accordingly steps to the right. Conversely, during backtracking, the state transitions via the failure pointer, effectively discarding any irrelevant left-side components. Note that in actual implementations, the failure links in AC automata are primarily used during the construction phase and the match phase. Once the automaton is constructed, these failure links are often replaced by virtual transitions that directly lead to the correct states like Algorithm 2. This optimization streamlines the matching process, enhancing efficiency by reducing unnecessary transitions.

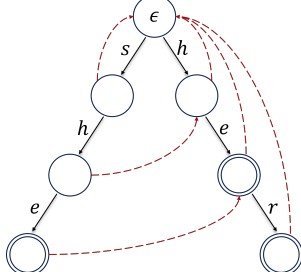

Figure 8: The illustration of an Aho–Corasick automaton with patterns "she", "he", and "her", with final states in double circles and failure links in red.

### D.3 LRU EVICTION STRATEGY

In RACER's retrieval automaton, we adopt an LRU (Least Recently Used) eviction mechanism to manage limited node capacity. When the maximum number of nodes is reached, the least recently accessed node is recycled and reassigned to represent a new state. This ensures that the automaton

---

**Algorithm 2** Calculate failure links for nodes in an AC automaton

---

1: **function** GET_TRANSITIONS
2:     Initialize an empty queue $Q$
3:     **for** $v \leftarrow$ children of the root **do**
4:         fail$(v) \leftarrow$ root                                   ▷ Set initial failure state to root
5:         Enqueue $v$ into $Q$
6:     **end for**
7:     **while** not $Q$.isEmpty() **do**
8:         $u \leftarrow Q$.dequeue()
9:         **for** $i \leftarrow$ possible transitions from $u$ **do**
10:            $v \leftarrow$ child$(u, i)$
11:            **if** $v \neq \epsilon$ **then**
12:                fail$(v) \leftarrow$ child(fail$(u), i)$                 ▷ Update the failure pointer
13:                Enqueue $v$ into $Q$
14:            **else**
15:                $f \leftarrow$ fail$(u)$
16:                **while** $f \neq$ root **and** child$(f, i) = \epsilon$ **do**
17:                    $f \leftarrow$ fail$(f)$                        ▷ Backtrack through the failure pointer
18:                **end while**
19:                **if** child$(f, i) \neq \epsilon$ **then**
20:                    fail$(v) \leftarrow$ child$(f, i)$
21:                **else**
22:                    fail$(v) \leftarrow$ root
23:                **end if**
24:            **end if**
25:        **end for**
26:    **end while**
27: **end function**

---

continuously adapts to the most relevant $n$-grams from the current decoding context while maintaining bounded memory.

The mechanism is shown in Algorithm 3 and works as follows: **Touch**: Every time a node is visited, it is moved to the front of the LRU list and its reference is updated in the hash map. This guarantees that the tail of the list always contains the least recently used node. **TouchPrefix**: When a failure transition occurs, not only the current node but also its ancestors along the prefix are "touched". This ensures that the entire matching path is marked as recently used. **TransTokens**: When processing a sequence of tokens, the automaton repeatedly performs *Touch* and failure transitions until either a matching child node is found or the traversal falls back to the root. **InsertTokens**: When inserting a new $n$-gram, if no free node is available, the node at the back of the LRU list (the least recently used one) is evicted and reset. It is then reassigned as the child for the new transition. Frequency counters along the insertion path are updated accordingly. This design ensures that: Only leaf nodes are evicted, preventing structural corruption of the automaton. The automaton remains adaptive to changing contexts, exploiting temporal and spatial locality during decoding. The eviction and update operations remain lightweight and efficient, keeping inference fast.

**Time Complexity** The Touch operation runs in constant time $\mathcal{O}(1)$, as it only updates the doubly linked list and hash map. TouchPrefix has a worst-case complexity of $O(d)$, where $d$ is the maximum depth of the automaton (i.e., the $n$-gram length, typically a small constant). The TransTokens procedure processes each token in the input sequence and may backtrack up to depth $d$ through failure links, giving a worst-case complexity of $\mathcal{O}(|\text{tokens}| \cdot d)$, though in practice it is close to linear $O(|\text{tokens}|)$. Finally, InsertTokens requires at most $d$ steps for each token sequence, resulting in $O(|\text{tokens}|)$ complexity.

**Space Complexity** The storage requirement is linear in the number of automaton nodes. Trie nodes occupy $O(|\mathcal{A}|)$ space, bounded by the maximum capacity of the automaton. The LRU list and hash map also require $O(|\mathcal{A}|)$ space, as each node is tracked in both structures. Thus, total

---

**Algorithm 3** LRU Eviction Strategy in AC Automaton

---

1: **function** TOUCH(node)
2:     Move node to the front of LRU_LIST
3:     Update LRU_MAP[node] ← LRU_LIST.begin()
4: **end function**
5: **function** TOUCHPREFIX(node)
6:     **while** node $\neq \epsilon$ **do**
7:         fail(node) ← root                          ▷ Default failure link to the root before rebuild
8:         TOUCH(node)
9:         node ← parent(node)
10:     **end while**
11: **end function**
12: **function** TRANSTOKENS(tokens)
13:     $u \leftarrow$ curren_state
14:     **for** each $t \in$ tokens **do**
15:         TOUCH($u$)
16:         **while** $u \neq$ root **and** child($u, t$) $= \epsilon$ **do**
17:             $u \leftarrow$ fail($u$)                              ▷ Failure transition
18:         **end while**
19:         TOUCHPREFIX($u$)                    ▷ Update prefix after failure transition
20:         **if** child($u, t$) $\neq \epsilon$ **then**
21:             $u \leftarrow$ child($u, t$)
22:         **else**
23:             $u \leftarrow$ root
24:         **end if**
25:     **end for**
26:     TOUCH($u$)                              ▷ Final state after processing tokens
27:     current_state ← $u$
28: **end function**
29: **function** INSERTTOKENS(tokens, frequency)
30:     $u \leftarrow$ root
31:     freq($u$) ← freq($u$) + frequency
32:     **for** each $t \in$ tokens **do**
33:         TOUCH($u$)
34:         **if** child($u, t$) $= \epsilon$ **then**
35:             new_node ← LRU_LIST.back()
36:             new_node.reset()
37:             child($u, t$) ← new_node
38:         **end if**
39:         $u \leftarrow$ child($u, t$)
40:         freq($u$) ← freq($u$) + frequency
41:     **end for**
42:     TOUCH($u$)                              ▷ Touch the leaf node
43: **end function**

---

memory is $O(|\mathcal{A}|)$, capped in practice at $10^4$-$10^5$ nodes, which corresponds to tens of megabytes – well within the budget of modern devices and suitable for memory-constrained scenarios.

### D.4    VERIFICATION OF AC AUTOMATA WITH LRU EVICTION

In the full RACER method, the failure links of AC automaton are constructed once during prefill because the retrieval component works together with the Logits Tree, and frequent updates bring negligible benefit. However, for the retrieval-only ablation, the retrieval component must operate independently. To fairly evaluate the retrieval structure itself (without support from the logits branch), we adopt a fixed-interval update of the automaton (every $K$ steps). This ensures that the retrieval-only version reflects a reasonable and competitive retrieval behavior, enabling a fair comparison with other retrieval-based baselines.

Table 9 reports the retrieval-only evaluation results across Spec-Bench, HumanEval, and MGSM-ZH using an RTX 3090 GPU, comparing PLD, REST, SAM-Decoding (SAMD), and the retrieval-only configuration of RACER. Overall, RACER consistently achieves the highest MAT scores and speedup ratios across all benchmarks. For the Vicuna-7B backbone, RACER surpasses the strongest baseline by a large margin: MAT improves by 0.20 to 0.70 across individual benchmarks, and its average MAT reaches 2.53, compared to 1.98 for SAMD and 1.95 for PLD. The speedup gains follow a similar pattern, with RACER achieving an average $2.05\times$ acceleration, exceeding all other retrieval-only approaches. The improvements are particularly pronounced on MGSM-ZH, where RACER attains a MAT of 3.32, significantly outperforming the baselines (1.29-2.65). Such gains indicate that RACER's retrieval mechanism exhibits stronger robustness under multi-step reasoning tasks, especially those involving localized patterns and longer dependency chains. Spec-Bench and HumanEval also show consistent benefits, demonstrating that RACER's retrieval operations are effective across both synthetic and code-generation workloads. Importantly, although RACER in this ablation does not leverage the Logits Tree and only employs the AC automaton for seen information, it still substantially outperforms other retrieval-only systems. This suggests that RACER's retrieval automaton is intrinsically more efficient and semantically aligned with the model's generation dynamics.

Table 9: Evaluation results on Spec-Bench, HumanEval and MGSM-ZH using an RTX 3090 GPU, compared with retrieval-only methods PLD, REST and SAM-Decoding (SAMD). $\star$ indicates that RACER here only employs the retrieval automaton for **seen information**, configured with 10,000 nodes, an $n$-gram length of 8, and updating AC automaton failure links every 20 steps.

| Models | Method | Spec-Bench | | HumanEval | | MGSM-ZH | | Average | |
|---|---|---|---|---|---|---|---|---|---|
| | | MAT | Speedup | MAT | Speedup | MAT | Speedup | MAT | Speedup |
| | PLD | 1.72 | 1.57 | 1.57 | 1.45 | 2.57 | 2.35 | 1.95 | 1.79 |
| | REST | 1.82 | 1.37 | 2.06 | 1.65 | 1.29 | 1.06 | 1.49 | 1.36 |
| Vicuna 7B | SAMD | 1.70 | 1.65 | 1.59 | 1.54 | 2.65 | 2.49 | 1.98 | 1.89 |
| | **RACER$^\star$** | **2.02** | **1.69** | **2.25** | **1.86** | **3.32** | **2.61** | **2.53** | **2.05** |

### E    SUPPLEMENTARY PRELIMENARY RESULTS

As shown in Figure 9, the supplementary experiments confirm the same trend observed in the main text (Figure 1 and Figure 2). Specifically, the *copy-logit* expansion consistently produces a sharper, heavy-tailed acceptance distribution compared to *last-logit*, with most accepted tokens concentrated in top ranks. Moreover, deeper $k$-ary expansions preserve this concentration and further validate the breadth allocation rule. These results demonstrate that the phenomena discussed in the main text are robust across different model scales and settings, reinforcing our choice of *copy-logit* as the default expansion strategy in RACER.

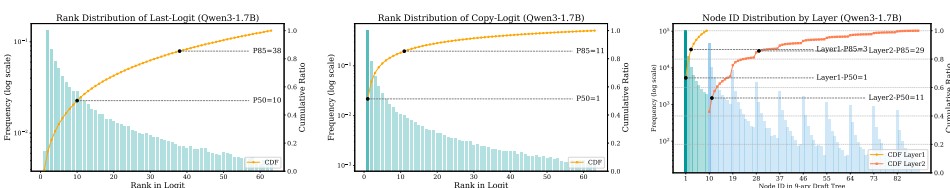

(a) Logits Tree with one layer be- (b) Logits Tree with one layer (c) Logits Tree with two lay-
yond next-token expanded with beyond next-token expanded ers beyond next-token expanded
*last-logit*. with *copy-logit*. with *copy-logit* in 9-ary manner.

Figure 9: Accepted draft statistics of Qwen3-1.7B on Spec-Bench.

# F COMPARISON WITH EAGLE-3

As a model-free method, RACER is orthogonal to model-based methods like EAGLE-3. While
EAGLE-3 benefits from extensive training and model-based optimizations, RACER provides sig-
nificant acceleration without requiring any additional training. Here, we explore in which scenarios
RACER can complement EAGLE-3, potentially providing better acceleration in a hybrid setting
with minimal modification to EAGLE-3.

In the supplementary results with Qwen3, we include several reasoning tasks: GSM8K (Cobbe
et al., 2021), AIME (Veeraboina, 2023), and MATH (Hendrycks et al., 2021), using AngelSlim's
re-implementation of EAGLE-3 model weights. We select 250 (+2) samples from each dataset: the
first 250 samples for GSM8K, 250 random samples for AIME (from 1983 to 2024), and 50 random
samples across levels 1 to 5, and 2 samples from level ? for MATH.

Table 10: Results on reasoning tasks: GSM8K, AIME, and MATH, reported in mean accepted
tokens (MAT) and speedup ratio. † denotes that the EAGLE-3 model weight is from AngelSlim's
re-implementation.

| Models | Method | GSM8K | | AIME | | MATH | | Average | |
|---|---|---|---|---|---|---|---|---|---|
| | | MAT | Speedup | MAT | Speedup | MAT | Speedup | MAT | Speedup |
| Qwen3 8B | EAGLE-3† | **3.86** | 2.65 | **3.44** | 2.44 | **3.55** | **2.60** | 3.62 | 2.56 |
| | RACER | 3.01 | **2.68** | 2.91 | **2.63** | 2.88 | 2.58 | 2.93 | **2.63** |
| Qwen3 14B | EAGLE-3† | **3.08** | 2.23 | **3.05** | 2.24 | **3.06** | 2.23 | **3.06** | 2.23 |
| | RACER | 2.95 | **2.72** | 2.90 | **2.64** | 2.87 | **2.55** | 2.91 | **2.64** |
| Qwen3 32B | EAGLE-3† | **3.32** | 2.51 | **3.26** | **2.34** | **3.33** | **2.42** | **3.30** | **2.42** |
| | RACER | 2.87 | **2.53** | 2.84 | 2.30 | 2.82 | 2.32 | 2.84 | 2.38 |

Table 10 shows that EAGLE-3 achieves higher MAT across all settings, which is expected given
its task-specific training and model-based predictive capability. However, RACER attains compara-
ble or even higher speedups in several cases (e.g., Qwen3-8B on GSM8K/AIME and Qwen3-14B
across all tasks), despite having lower MAT. This highlights a key advantage of model-free decod-
ing: RACER incurs nearly constant draft-generation cost, so its speedup does not degrade as sharply
as computation-heavy model-based methods when model size increases. Across Qwen3-14B and
Qwen3-32B, RACER maintains stable acceleration (2.38-2.72×), while EAGLE-3's speedup re-
mains limited by verification overhead. These results suggest that RACER's speculative drafting is
highly efficient even for long reasoning tasks, and that its acceleration stems from computational
structure rather than model training. Most importantly, this comparison illustrates that RACER and
EAGLE-3 offer orthogonal strengths: EAGLE-3 excels when high-quality next-step predictions are
available via training. RACER excels when low overhead and robustness across tasks and model
sizes are required.

Notably, even with the official Vicuna-13B-v1.3 EAGLE-3 weight – trained on substantially broader
and more diverse data – the gap between EAGLE-3 and RACER on MGSM-ZH remains relatively

Table 11: Results on Spec-Bench, HumanEval, and MGSM-ZH, reported in mean accepted tokens (MAT) and speedup ratio.

| Models | Method | Spec-Bench | | HumanEval | | MGSM-ZH | | Average | |
|---|---|---|---|---|---|---|---|---|---|
| | | MAT | Speedup | MAT | Speedup | MAT | Speedup | MAT | Speedup |
| Vicuna 13B | EAGLE-3 | 5.70 | 3.36 | 6.41 | 3.92 | 4.69 | 2.90 | 5.60 | 3.39 |
| | RACER | 2.92 | 2.23 | 3.22 | 2.50 | 3.57 | 2.79 | 3.24 | 2.51 |

small (Table 11). This suggests that RACER retains competitive efficiency even in settings where the model-based draft generator has seen relevant training data.

SAM Decoding surpasses EAGLE-2 once combined with it, indicating that a lightweight model-free router can further improve well-trained model-based systems. By analogy, integrating RACER with EAGLE-3 is a natural and promising direction: RACER can provide efficient multi-branch drafting while EAGLE-3 provides high-quality predictions. We chose not to include this hybrid variant in the main submission to keep the method lightweight and implementation minimal, but we believe it represents a valuable avenue for future work.

## G  USE OF LARGE LANGUAGE MODELS

In this work, Large Language Models (LLMs) were employed solely to provide assistance with language editing and textual clarification during the preparation of this paper. All technical ideas, methodological designs, analyses, and experimental studies were conceived, executed, and validated exclusively by the human authors.

