# OpenReview forum: "RACER: Retrieval-Augmented Contextual Rapid Speculative Decoding"
_ICLR.cc/2026/Conference — ICLR 2026 Conference Withdrawn Submission_

### Official Review · Reviewer_M1gZ · 2025-10-16

**Soundness:** 3
**Presentation:** 3
**Contribution:** 2
**Rating:** 4
**Confidence:** 5

**Summary:**

This paper proposes RACER (Retrieval-Augmented Contextual Rapid Speculative Decoding), a lightweight
and training-free framework that integrates retrieved exact patterns with logit-driven future cues. Experiments on Spec-Bench, HumanEval, and MGSM demonstrate that RACER consistently accelerates inference and can achieve 2.2~2.8x speedup.

**Strengths:**

This paper is technically sound and easy to understand.

The experimental results show the effectiveness of the proposed method.

**Weaknesses:**

The paper focuses on generating tree structure and improve the overall MAT to speedup the large language model.

However, they only conduct experiments with HuggingFace Transformers framework. Here comes a problem that the method may not have such speedup on the popular inference framework such as vLLM. In fact, HuggingFace Transformers framework does not optimize the speed of LLMs very well, which makes the ratio of the latency of tree generation process smaller. When using vLLM framework where the operations in LLMs are optimized very well, the tree generation process will take more time and reduce the speedup.

The author should verify their method on such inference frameworks to show that their method is actually useful in reality.

**Questions:**

See weaknesses above.

---

> ### Author Response · Authors · 2025-11-20
> **Response to Reviewer M1gZ**
>
> We appreciate the reviewer's concern. Our experiments are conducted on Hugginface Transformers (HF) primarily because it is easier to modify for academic prototyping; speculative decoding methods are almost always first validated in HF before being ported to production systems, like EAGLE and PLD (named `ngram` in vLLM/SGLang).
>
> RACER's draft-generation overhead is extremely small and remains nearly constant as the sequence grows. Importantly, the MAT (Mean Accepted Tokens) is largely independent of the inference backend. Across all tasks, RACER maintains MAT $\approx 3$, which suggests similar speedup behavior under optimized frameworks.
>
> As shown in the model-scale ablations discussed in our response to Reviewer 9qLi, RACER's MAT remains almost constant even as model size increases. Because RACER is model-free, its draft-generation cost is stable, whereas model-based approaches incur higher verification cost for larger models. This gives RACER a relative advantage as model size grows in the real production environment. **We are trying to integrate RACER into vLLM/SGLang.**
>
> Here's the model size ablation results:
>
> | Qwen3 | 0.6B | 1.7B | 4B | 8B | 14B | 32B |
> | :-: | :-: | :-: | :-: | :-: | :-: | :-: |
> | MAT | 2.89 | 2.78 | 2.78 | 2.73 | 2.67 | 2.66 |
> | Speedup | 2.55 | 2.48 | 2.53 | 2.13 | 2.23 | 2.17 |
>
> ---
>
> We believe these explanations address your concerns and clarify the contributions of our method. We appreciate your thoughtful review and hope this response positively informs your final evaluation.

---

> ### Author Response · Authors · 2025-11-27
>
> Dear Reviewer,
>
> I hope this message finds you well. As the discussion period is nearing its end, we wanted to follow up to ensure that all of your earlier concerns have been fully addressed. In our latest round of revisions, some of the points raised in the discussion have been clarified and have received positive feedback, indicating that the updates effectively addressed those earlier concerns. We hope that these clarifications may also speak to the points you previously highlighted. If there is anything further you feel would benefit from additional explanation or refinement, please feel free to let us know. We would truly appreciate any further feedback you may have.
>
> Thank you again for your time and effort in reviewing our paper.

---

### Official Review · Reviewer_JwCs · 2025-10-20

**Soundness:** 2
**Presentation:** 1
**Contribution:** 2
**Rating:** 2
**Confidence:** 4

**Summary:**

This paper proposes RACER, a training-free decoding acceleration framework that combines two types of speculative signals: (1) a logits tree constructed using a copy-logits strategy to extrapolate high-probability tokens, and (2) a retrieval tree maintained by an AC automaton to leverage repeated patterns in the context. These two sources are integrated into a unified speculative draft, which is then verified using standard speculative decoding. The paper reports empirical results on several benchmarks, showing that RACER achieves a 2.2-2.8x speedup over standard autoregressive decoding and outperforms other training-free baselines such as TokenRecycling and LogitSpec.

**Strengths:**

1. The paper addresses an important and practical problem: accelerating LLM decoding without additional training. This is highly relevant for both research and deployment.

2. The proposed method integrates retrieval-based and logits-based drafting in a unified framework, which is conceptually simple and compatible with existing speculative decoding systems.

3. The method achieves consistent speedups across tasks and shows robustness with respect to hyperparameters such as draft size, retrieval tree capacity, and n-gram depth.

**Weaknesses:**

1. **The paper is poorly written.** For example, the main contributions are the use of copy-logit for building the logits tree and the use of an AC automaton for maintaining the retrieval tree. However, the entire article uses only one sentence to explain “copy-logit”, and that sentence is not understandable (see Weakness 2). In addition, the acronym “LRU” is explained only the third time it appears, which is not reader-friendly. Moreover, **there are many very short paragraphs with only 2 or 3 lines** (e.g., lines 162–167 and 216–222), which makes the paper look unprofessional. There is also a missing period at the end of line 185.

2. The definitions of the last-logit and copy-logit strategies are unclear. In particular, the meaning of “the same token” in line 125 is ambiguous; the paper should specify what the token is “the same as”. It is strongly recommended to explain these concepts more clearly. In Figure 1, the differences between white and green circles, as well as between solid and dashed arrows, should be explicitly explained in the figure caption.

3. The node index definition in lines 165–167 is non-standard. Using proper mathematical symbols would make it clearer and more precise.

4. The choice of the MGSM-ZH dataset is somewhat questionable. From Table 1, it seems this dataset was chosen mainly because EAGLE-3 performs poorly on it, allowing RACER to surpass EAGLE-3 on average. However, the reproduced EAGLE-3 model was not trained on Chinese data, making this comparison problematic. If my guess is true, I don't think there's any need to be so secretive. It's normal for retrieval-based methods to not outperform EAGLE-3, and it would be strange if they could.

5. The novelty is somewhat weak. The authors claim three contributions. The copy-logit strategy is not clearly explained. The retrieval tree with AC automaton is acceptable, but the integration is rather trivial.

**Questions:**

1. How do copy-logit and last-logit work? In Figure 1, what is the difference between the white circles and the green ones? What is the difference between a solid arrow and a dashed arrow?

2. What is b in Equation 3? Is it the same as k in a k-ary tree?

3. Why is the experiment done on MGSM-ZH, which is not popular in reasoning tasks? It is advised to do the experiments on more popular datasets, such as the original dataset of MGSM-ZH, GSM8K, or some other popular datasets such as AIME and AMC.

---

> ### Author Response · Authors · 2025-11-20
> **Response to Reviewer JwCs [1/2]**
>
> **On Writing Clarity and Missing Definitions**
>
> The clarity concerns raised were focused on specific definitions and figure explanations. At the same time, other reviewers noted that the overall methodology was clearly presented. We have now improved the problematic parts and standardized terminology across the paper.
> 1. Copy-logit vs. last-logit
> We expanded the definition, clarified the meaning of "the same token" in Section 3.1, and added a case study in Figure 6 in Appendix D.1 illustrating the Logits Tree construction. Figure 1 now explicitly explains the meaning of node colors and solid/dashed arrows.
> 2. LRU definition
> LRU (Least Recently Used) is now introduced upon first occurrence rather than on the third mention (Lines 116-118).
> 3. Indexing conventions
> The node indexing in Lines 165–167 has been rewritten using proper mathematical notation $\text{parent}(i) = \lfloor\frac{i - 1}{k}\rfloor$, and a visual example is added to the Figure 6 in Appendix D.1.
> 4. Paragraph organization
> Several short paragraphs have been merged and symbols standardized to improve readability.
>
> **On the Meaning of $b_i$ in Equation 3**
>
> We clarified that $b_i$ denotes the *breadth* (number of children) of node  $i$. Specifically:
> - the root has $b_0 = k$;
> - first-layer nodes receive $b_1 = b_0, b_2=b_0/2, b_3=b_0/4, \dots$;
> - deeper nodes inherit at most half of their parent's breadth (e.g., $b_{k + 1} = b_1/2, b_{k + 2} = b_1/4, \dots$).
>
> This schedule encourages broad exploration near the top while allowing controlled deepening, striking a balance between exploring long-tail continuations and preserving MAT.

---

> > ### Author Response · Authors · 2025-11-20
> > **Response to Reviewer JwCs [2/2]**
> >
> > **On the Choice of MGSM-ZH**
> >
> > We clarified our dataset choices:
> > - Spec-Bench already includes GSM8K as the reasoning category.
> > - We selected MGSM-ZH to evaluate multilingual generalization, following PEARL (ICLR’25).
> > - We did not include all MGSM languages because Vicuna shows unstable generation quality for several languages, which would introduce confounding effects. Chinese (especially for Qwen3) is a well-covered training language and provides a fair multilingual testbed.
> >
> > To address the reviewer's suggestion, we additionally evaluated **GSM8K, AIME, and MATH** (Appendix Table 10). Here we post average results for a quick check:
> >
> > | Method | Qwen3-8B MAT | Qwen3-8B Speedup | Qwen3-14B MAT | Qwen3-14B Speedup | Qwen3-32B MAT | Qwen3-32B Speedup |
> > | :-: | :-: | :-: | :-: | :-: | :-: | :-: |
> > | EAGLE-3$^\dagger$ | **3.62** | 2.56 | **3.06** | 2.23 | **3.30** | **2.42** |
> > | RACER | 2.93 | **2.63** | 2.91 | **2.64** | 2.84 | 2.38 |
> >
> > > $\dagger$ denotes EAGLE-3 model weights come from AngelSlim's re-implementation.
> >
> > In Table 1, **RACER holds up strongly even without MGSM-ZH**:
> >
> > * Qwen3-14B: RACER > EAGLE-3
> > * Qwen3-8B: $2.19\times$ vs. $2.29\times$
> > * Qwen3-32B: $2.13\times$ vs. $2.15\times$
> >
> > These results highlight RACER's **model-free advantages in generalization and transferability**, especially when verification cost grows with model size.
> >
> > Finally, we emphasize that **RACER and EAGLE-3 are orthogonal**. RACER is model-free and can be combined with model-based approaches; we are currently developing a **hybrid** version.
> >
> > ---
> >
> > We believe these explanations address your concerns and clarify the contributions of our method. We appreciate your thoughtful review and hope this response positively informs your final evaluation.

---

> ### Comment · Reviewer_JwCs · 2025-11-21
>
> Thank you for the detailed rebuttal and for taking the time to revise the paper. I would raise my score from 2 to 6.
>
> Firstly, I do want to note that the sentence “other reviewers noted that the overall methodology was clearly presented” felt a bit surprising. Two out of the four reviewers explicitly mentioned issues with the writing, so I personally would hesitate to describe the original version as clearly written. That said, the revised manuscript is indeed noticeably more readable, and the improvements in paragraph structure, terminology consistency, and figure explanations have made a meaningful difference. I appreciate the effort the authors put into polishing these aspects.
>
> Regarding the distinction between copy-logit and last-logit, the expanded explanation was helpful. I now better understand what “copy-logit” refers to, and I find the idea quite interesting. If possible, I would suggest adding a small legend or clarification in the caption directly within Figure 1, so that readers can immediately interpret the colors and arrows without jumping to the previous sections.
>
> The authors’ reasoning for selecting MGSM-ZH is also reasonable. Given the instability in multilingual generation for Vicuna and the strong coverage of Chinese for Qwen3, the choice becomes clearer. I think adding a brief explanation to the paper would be beneficial, since it only takes a sentence or two to give readers the necessary context. I also appreciate the additional AIME and MATH results provided in the rebuttal. While they are not fully exhaustive, this is understandable under rebuttal time constraints. For the final version (if the paper is accepted), it would strengthen the paper to complete the remaining evaluations if feasible, especially since AIME and MATH are relatively small and directly relevant to the paper’s claims.
>
> Lastly, I have a couple of small formatting suggestions that might help with the overall presentation. There seems to be some inconsistency in the use of hyphens versus em dashes. For example, Line 204 uses “-” while Line 214 uses “—”. A quick search-and-replace should make this consistent. Also, the table on Page 8 looks unusually large in font size compared to the surrounding text. If this were done using `\resizebox`, it might be worth considering a normal-sized table to avoid the visual jump.
>
> Overall, I appreciate the authors’ thorough responses and the significant improvements made during the rebuttal. The clarifications and updates address my main concerns, and with a bit more polish, the paper will be in much stronger shape.

---

> > ### Author Response · Authors · 2025-11-22
> > **Response to Reviewer JwCs**
> >
> > Thank you very much for the helpful follow-up comments and for re-evaluating the submission.
> >
> > Based on your suggestions, we further improved the clarity of the paper:
> >
> > - **Clarification of Figure 1**: We added clarification to Figure 1 to explicitly explain what the nodes and dashed edges represent and how the _guess-and-verify_ pipeline works (Lines 142-147).
> > - **Explanation of MGSM-ZH**: We added a brief justification for choosing MGSM-ZH to provide more background and context to readers (Lines 320-323).
> > - **Formatting Issues**: We standardized the use of hyphens and dashes throughout the paper and unified the font sizes in tables to ensure visual consistency.
> >
> > We sincerely appreciate your feedback, which has significantly helped us improve the readability of the paper. If there are any other parts you feel could be further improved, we would be very happy to revise them.
> >
> > Thank you again for your time and constructive comments.

---

### Official Review · Reviewer_k6pL · 2025-10-31

**Soundness:** 2
**Presentation:** 1
**Contribution:** 2
**Rating:** 2
**Confidence:** 4

**Summary:**

This paper introduces RACER, a training‑free framework that integrates retrieval-based speculative decoding (SD) with logit‑based drafts. Experiments on Spec-Bench, HumanEval, and MGSM demonstrate that RACER consistently accelerates inference, achieving a speedup of compared to autoregressive decoding, and outperforms prior training-free methods, offering a scalable, plug-and-play solution for efficient LLM decoding.

**Strengths:**

1. This work presents a valuable exploration to integrate retrieval-based SD methods with logit-based drafting. It provides an insightful idea that SD should not only leverage retrieval contents, which provide seen information through exact pattern matches, but also logit-based contents, which supply unseen information for future tokens.
2. The proposed method is assessed on a wide range of text generation benchmarks, including Spec-Bench, HumanEval, and MGSM. The evaluated models cover the Vicuna series and Qwen3 series.
3. RACER consistently accelerates inference, achieving a speedup of 2.2x-2.8x compared to autoregressive decoding, and outperforms prior training-free methods, such as token recycling, logitspec, PLD, and REST.

**Weaknesses:**

1. **Comparison with LogitSpec**: A similar idea of combining retrieval-based SD with logit-based drafts has been proposed in LogitSpec [1], which appears to be the most closely related work to RACER. A more thorough comparison between LogitSpec and RACER is needed to better highlight this paper’s contributions.
2. **Writing Refinement**: The manuscript would benefit from significant improvements in writing clarity. For example, Section 3.1 fails to clearly explain the “copy-logit strategy.” Although I carefully reviewed Lines 124–126 and Figure 1, the explanation lacks sufficient detail. Given that the copy-logit strategy is a central contribution of the logit-based drafts, the lack of a clear and detailed description creates confusion for the reader.
3. **Insufficient Methodological Details**: (1) In Figure 1, the relationship between frequency (y-axis) and the Mean Accepted Tokens (MAT) is unclear. (2) The term “failure link” in Lines 213–214 is not well defined. (3) Section 3.2 omits several critical details. For instance, what specific retrieval method is employed? Additionally, the update rule described in Lines 216–219 is presented too briefly, without adequate explanation.
4. **Lack of Experimental Clarity**: It is unclear what the maximum number of draft tokens is for each SD step. Among these draft tokens, what is the proportion of retrieval tokens versus logit tokens? It is also confusing that Line 425 states the draft size ranges from 16 to 64, while the maximum breadth of the logit tree is 8, and the retrieval tree contains up to 10,000 nodes with an n-gram length of 10. The relationships among these parameters should be clearly clarified.
5. **Missing Baseline Comparison**: Why is RACER not compared against EAGLE-3 using the Vicuna series?
6. Table 2 illustrates that the logit-based drafting contributes more to the efficiency of RACER, while the improvement from the retrieval part seems to be much smaller. The contributions and necessity of each designed module should be further analyzed.

**Questions:**

1. The font size in Figures 1 and 2 is too small and difficult to read.
2. The phrase “P50 and P85 quantiles” in Lines 148–149 could be revised to “50th and 85th percentiles” for clarity and consistency.
3. A grammar issue: the paragraph spanning Lines 183–186 ends without a period.


[1] LogitSpec: Accelerating Retrieval-based Speculative Decoding via Next Next Token Speculation. Liu et al. 2025.

---

> ### Author Response · Authors · 2025-11-20
> **Response to Reviewer k6pl [1/3]**
>
> **On Writing, Clarity, and Missing Definitions**
>
> The clarity concerns raised were focused on specific definitions and figure explanations. At the same time, other reviewers noted that the overall methodology was clearly presented. We have now improved the problematic parts and standardized terminology across the paper.
> 1. Figures, formatting, and typography
>     - Figures 1 and 2: We have increased the font size and replaced both figures with high-resolution versions for improved readability.
>     - Terminology in Section 3: "P50 / P85 quantiles" has been revised to "50th / 85th percentiles" (updated in Lines 152–155).
>     - Punctuation: The missing period at the end of the paragraph originally spanning Lines 183–186 has been fixed (now Lines 207–209).
> 2. Clarifying technical definitions and explanations
>    - Copy-logit strategy (Section 3.1): We significantly expanded the explanation of the copy-logit mechanism and added a concrete case study in the appendix to ensure the definition is precise and unambiguous.
>     - The y-axis "frequency" in Figure 1 is not functionally related to MAT; the two statistics are presented in parallel since MAT is a global metric and frequency is a relative statics used to analyze the rank distribution of accepted "first token beyond next token".
>     - Failure links: We have already included formal definitions and algorithms about AC automata and failure links in Appendix D.2. To be more reader friendly, we added the missing definition in the main text (Lines 216–224).
>     - Missing details in Section 3.2: We enriched this section with the specific retrieval and eviction mechanism, a clearer description of the AC automaton transitions, and the update rule.

---

> ### Author Response · Authors · 2025-11-20
> **Response to Reviewer k6pL [2/3]**
>
> **On Comparison with LogitSpec**
>
> 1. Logit information: last-logit vs. recursive copy-logit expansion
>    - LogitSpec relies on the last-logit signal to predict only the next–next token, and retrieves relevant reference for both the next token and the next next token.
>    - In contrast, RACER adopts a copy-logit mechanism, which recursively expands a Logits Tree across the top-$k$ candidates. Under this view, LogitSpec corresponds to using last-logit solely for generating the first child of the root node, while RACER constructs a multi-level tree that captures a richer distributional structure and yields significantly higher MAT and speedup raios.
> 2. Retrieval structure: hash-table storage vs. AC automaton with LRU eviction
>    - LogitSpec stores $n$-grams in a hash table, enabling constant-time lookup but with memory usage that grows with sequence length, which makes long-context inference increasingly expensive to maintain.
>     - RACER instead employs an Aho–Corasick automaton equipped with an *LRU-based eviction* mechanism. This design not only provides $\mathcal{O}(1)$ effective transition/match operations, but also *maintains bounded memory* even as the sequence grows. This makes RACER better suited for long-context decoding.
> 3. Re-evaluating the role of retrieval vs. logits
>    - LogitSpec characterizes retrieval-based signals as the dominant contributor to speedup and treats logits-based signals as auxiliary. However, our experiments consistently show that *logit-derived information is in fact the stronger driver of speculative acceptance*.
>    - This discrepancy arises from the choice of the logit signal: LogitSpec relies on the last-logit prediction, which provides only a shallow, single-step lookahead and therefore underestimates the predictive power of logits in speculative decoding.
>    - RACER uses copy-logit recursion to construct multi-level logit-driven drafts, revealing that logits carry substantially more robust forward information than previously observed.
>
> **On Experimental Clarity**
>
> 1. Draft size setting: The maximum draft size per speculative step is 64 for all experiments, except in ablation studies where it is explicitly varied. We updated this in Lines 307-308.
> 2. Retrieval vs. logit proportions.
> This ratio is task-dependent. Retrieval dominates in pattern-repetitive tasks (e.g., RAG-style), while logit-based drafts dominate in open-ended generation.
> 3. Relation among draft size, Logits Tree width, Retrieval Tree capacity and $n$-gram depth.
> We added clarification in our new version (Lines 463–489).

---

> ### Author Response · Authors · 2025-11-20
> **Response to Reviewer k6pL [3/3]**
>
> **On the Lack of EAGLE-3 Baseline with Vicuna**
>
> - For Vicuna series, there's only 13B model with EAGLE-3 weight, so we only adopt AngelSlim's re-implementation of Qwen3 series for a more comprehensive comparison, expanding from 8B to 32B. And we now add Vicuna-13B EAGLE-3 results in the Appendix and further discussion between EAGLE-3 and RACER.
> - We want to emphasize that the contribution of EAGLE-3 and our RACER are **orthogonal**. We add EAGLE-3 results as the representative of model-based method to reveal what senarios model-free RACER could aid model-based methods with lighter and faster drafting process. We are working on a **hybrid** version with EAGLE-3.
>
> **On Module-wise Contribution Analysis**
>
> - We agree that logit-based drafting plays the dominant role. This is expected: logits encode richer contextual information, and Table 2 already reflects this observation. In RACER, the Retrieval Tree is designed as a complementary module rather than the primary driver.
> - To clarify contribution of retrieval, we added a comprehensive comparison between retrieval-only RACER and other retrieval-only baselines such as PLD, REST, and SAM Decoding (SAMD) on Vicuna-7B (Table 9 in Appendix D.4). Retrieval-only RACER **consistently outperforms all these methods**, demonstrating that our LRU-regulated AC automaton provides a substantially more efficient and accurate retrieval mechanism, even when used alone. These results show that although logits dominate overall performance, the retrieval module contributes meaningfully and is not redundant.
>
> To give a quick check, here's the average results among Spec-Bench, HumanEval and MGSM-ZH of retrieval-only experiments:
>
> | Method | PLD | REST | SAMD | RACER$^\star$ |
> | :-: | :-: | :-: | :-: | :-: |
> | MAT | 1.95 | 1.49 | 1.98 | **2.53** |
> | Speeup | 1.79 | 1.36 | 1.89 | **2.05** |
>
> > $\star$ indicates that RACER here only employs the retrieval automaton for seen information, configured with 10,000 nodes, an n-gram length of 8, and updating AC automaton failure links every 20 steps.
>
> ---
>
> We believe these explanations address your concerns and clarify the contributions of our method. We appreciate your thoughtful review and hope this response positively informs your final evaluation.

---

> ### Author Response · Authors · 2025-11-27
>
> Dear Reviewer,
>
> I hope this message finds you well. As the discussion period is nearing its end, we wanted to follow up to ensure that all of your earlier concerns have been fully addressed. In our latest round of revisions, some of the points raised in the discussion have been clarified and have received positive feedback, indicating that the updates effectively addressed those earlier concerns. We hope that these clarifications may also speak to the points you previously highlighted. If there is anything further you feel would benefit from additional explanation or refinement, please feel free to let us know. We would truly appreciate any further feedback you may have.
>
> Thank you again for your time and effort in reviewing our paper.

---

### Official Review · Reviewer_9qLi · 2025-11-04

**Soundness:** 3
**Presentation:** 3
**Contribution:** 2
**Rating:** 6
**Confidence:** 2

**Summary:**

This paper unifies logit-based and retrieval-based training-free speculative decoding within a single framework. Experimental results demonstrate superior performance compared to either approach used independently.

**Strengths:**

1.	The paper unifies two subcategories of training-free speculative decoding, achieving a better trade-off between acceptance rate and inference efficiency.
2.	The methodology and presentation are clear and easy to follow, making the approach straightforward to reproduce.

**Weaknesses:**

1.	The experiments primarily focus on a batch size of 1, which limits the generality of the throughput evaluation and may not reflect performance in higher-throughput settings.
2.	The main contribution appears to be the integration of two existing methods, offering limited novelty and conceptual insight beyond their combination.

**Questions:**

Throughput scaling: Could you report results for batch sizes > 1 to assess how throughput and latency scale beyond the single-batch setting?

Model-size ablation: The current results suggest the method may benefit more on larger models. Since most evaluations are on small–to–medium scales, could you include an ablation across model sizes to quantify how effectiveness changes with model capacity?

---

> ### Author Response · Authors · 2025-11-20
> **Response to Reviewer 9qLi [1/2]**
>
> **On Throughput Scscalability**
>
> Current implementations of mainstream speculative decoding methods in PyTorch Transformers (e.g., Token Recycling (ACL'25) and SAM-Decoding (ACL'25)) are restricted to `batch_size = 1`. Supporting larger batch sizes requires substantial modifications to the underlying codebase. We are currently exploring extensions to support this setting.
>
> **On Model-scale Ablation**
>
> We provide additional ablations on the Qwen3 series (0.6B to 32B). The MAT remains nearly constant (2.66-2.89) across all model sizes, indicating that RACER's guessing probability does not depend on model scale. Since RACER is model-free, its draft-generation cost stays essentially constant, while verification becomes more expensive for larger models. This naturally leads to a mild drop in speedup for very large models, which is expected for all speculative decoding methods. This follows from the fact that verification dominates latency for larger models. Importantly, RACER still achieves $>2\times$ acceleration even on 32B model, showing that the method scales well. Given RACER's lightweight design, its degradation at batch size >1 should be less severe than model-based methods. For more discussion, please see Table 7 and 8 and discussion in Appendix C of updated version.
>
> | Qwen3 | 0.6B | 1.7B | 4B | 8B | 14B | 32B |
> | :-: | :-: | :-: | :-: | :-: | :-: | :-: |
> | MAT | 2.89 | 2.78 | 2.78 | 2.73 | 2.67 | 2.66 |
> | Speedup | 2.55 | 2.48 | 2.53 | 2.13 | 2.23 | 2.17 |

---

> ### Author Response · Authors · 2025-11-20
> **Response to Reviewer 9qLi [2/2]**
>
> **On "limited novelty / simple integration"**
>
> We respectfully note that RACER is **not a simple combination** of existing techniques.
> Our work introduces a **new conceptual formulation** of speculative drafting:
> - **retrieval signals** encode seen information grounded in the observed context
> - **logit signals** encode unseen information predicted by the model.
>
> This explicit separation which is absent in prior work motivates a unified design consisting of:
> - coordinated breadth allocation in the Logits Tree
> - an LRU-regulated AC automaton for retrieval maintenance
> - joint pruning under a shared draft-capacity budget
>
> These components arise from the conceptual framework rather than from straightforward combination, and together they produce the empirical improvements demonstrated in the experiments.
>
> ---
>
> We believe these explanations address your concerns and clarify the contributions of our method. We appreciate your thoughtful review and hope this response positively informs your final evaluation.

---

> ### Author Response · Authors · 2025-11-27
>
> Dear Reviewer,
>
> I hope this message finds you well. As the discussion period is nearing its end, we wanted to follow up to ensure that all of your earlier concerns have been fully addressed. In our latest round of revisions, some of the points raised in the discussion have been clarified and have received positive feedback, indicating that the updates effectively addressed those earlier concerns. We hope that these clarifications may also speak to the points you previously highlighted. If there is anything further you feel would benefit from additional explanation or refinement, please feel free to let us know. We would truly appreciate any further feedback you may have.
>
> Thank you again for your time and effort in reviewing our paper.

---

### Note · Authors · 2026-01-05

I have read and agree with the venue's withdrawal policy on behalf of myself and my co-authors.